# Unexpected functional implication of a stable succinimide in the structural stability of *Methanocaldococcus jannaschii* glutaminase

Sanjeev Kumar[1], Sunita Prakash[2], Kallol Gupta[2,†], Aparna Dongre[1], Padmanabhan Balaram[2] & Hemalatha Balaram[1]

Protein ageing is often mediated by the formation of succinimide intermediates. These short-lived intermediates derive from asparaginyl deamidation and aspartyl dehydration and are rapidly converted into β-aspartyl or D-aspartyl residues. Here we report the presence of a highly stable succinimide intermediate in the glutaminase subunit of GMP synthetase from the hyperthermophile *Methanocaldoccocus jannaschii*. By comparing the biophysical properties of the wild-type protein and of several mutants, we show that the presence of succinimide increases the structural stability of the glutaminase subunit. The protein bearing this modification in fact remains folded at 100 °C and in 8 M guanidinium chloride. Mutation of the residue following the reactive asparagine provides insight into the factors that contribute to the hydrolytic stability of the succinimide. Our findings suggest that sequences that stabilize succinimides from hydrolysis may be evolutionarily selected to confer extreme thermal stability.

[1] Molecular Biology and Genetics Unit, Jawaharlal Nehru Centre for Advanced Scientific Research, Jakkur, Bangalore 560064, India. [2] Molecular Biophysics Unit, Indian Institute of Science, Bangalore 560012, India. † Present address: Department of Chemistry, Physical and Theoretical Chemistry Laboratory, University of Oxford, South Parks Road, Oxford OX1 3QZ, UK. Correspondence and requests for materials should be addressed to H.B. (email: hb@jncasr.ac.in).

A commonly occurring, spontaneous post-translational modification in proteins, both *in vitro* and *in vivo*, is asparaginyl deamidation and aspartyl isomerization to β-aspartyl residues[1–5]. Aspartyl isomerization and in most cases, asparaginyl deamidation proceeds through the succinimidyl intermediate[2–6]. Most succinimides in proteins are unstable, being prone to rapid hydrolysis resulting in their conversion to aspartyl (Asp/D) and isoaspartyl (isoAsp/isoD) residues, where formation of isoAsp arises from the transfer of the peptide backbone from α-carbonyl of Asp/Asn to the side chain β-carboxyl[2,3] (Fig. 1a). The succinimide is also prone to racemization due to increased acidity of the α-carbon[7,8] and hydrolysis of D-succinimide results in inversion of configuration to yield D-Asp and D-isoAsp. Further, succinimide formation through nucleophilic attack of side chain amide nitrogen on the peptide backbone carbonyl results in peptide chain cleavage[2,9]. All these changes, associated with the succinimide intermediate result in destabilization of protein structure and loss of function leading to cellular and physiological defects[2–4,9–12]. Asparaginyl deamidation and aspartyl isomerization through succinimidyl intermediates have been an impediment to the therapeutic use of recombinant proteins, a notable example being monoclonal antibodies that lose function on storage due to this post-translational alteration[13].

Despite the widely accepted loss of function associated with deamidation, several studies indicated that deamidation might have a biological significance[1,14–17]. It has been suggested that the process may serve as a molecular timer of biological events and also as a route for the post-translational production of unique proteins, *in vivo*[1,14–17]. Studies on the mechanism of deamidation/dehydration of Asn/Asp residues in proteins indicate that succinimide is metastable and the only functional role attributed to this intermediate is in protein splicing[18,19]. This apart, no other evidence for a biochemical or structural role has been associated with the succinimidyl intermediate.

The succinimide resulting from the cyclization of the side chain of Asn/Asp by imide formation with the backbone nitrogen of the succeeding residue should reduce the backbone conformational flexibility of the polypeptide segment. Reduced conformational flexibility is exploited by proteins in hyperthermophiles to retain their structure and function at high temperatures. Thermophilic enzymes generally possess increased number of ionic, hydrogen bonding, hydrophobic and van der Waals interactions along with tighter core packing, enhanced number of secondary structures and reduced length of surface loops to maintain structural stability at elevated temperatures[20–23]. Further, approaches employed to enhance thermal stability of mesophilic proteins involve structural alterations that restrict conformational flexibility and these include the introduction of disulfide bridges, replacement of glycyl residues with L-amino acids and mutation of loop residues to proline[24–27]. Similarly, one would expect that the presence of succinimide, if protected from hydrolysis, would reduce backbone conformational freedom and impart rigidity to the structure. Here we report on the role played by the post-translational modification of a specific asparaginyl residue, N109 to succinimide in *Methanocaldococcus jannaschii* glutaminase (MjGATase) in imparting structural stability to the enzyme. Our studies show that the succinimide in this enzyme is remarkably stable and resists hydrolysis even at 100 °C and high concentrations of chaotrope. Further, we demonstrate that the succinimide is indispensable for the structural stability and therefore, the enzymatic activity of MjGATase at elevated temperatures.

## Results

**N109 in MjGATase is modified to succinimide.** The electrospray ionization mass spectral (ESI-MS) analysis of several batches of both freshly purified and stored enzyme samples yielded a protein molecular mass of 21,003 Da (Fig. 1b), which is 17 Da lower than the predicted mass of 21,020 Da, suggesting the possibility of loss of an $NH_3$ molecule from one of the asparaginyl or glutaminyl residues in the protein. As purification of MjGATase involved a step of heating and anion-exchange chromatography, an N-terminal $(His)_6$-tagged MjGATase was generated and the recombinant protein was directly purified from the cell lysate using Ni-NTA affinity chromatography to rule out artefacts arising from the purification procedure. Fractions containing MjGATase, when pooled and examined by ESI-MS also showed loss of 17 Da (Supplementary Fig. 1), suggesting that the anomaly in the observed mass is not a consequence of the purification procedure. As MjGATase, in solution under native conditions is completely resistant to proteolysis by trypsin, in-gel trypsin digestion was carried out to identify the site of $NH_3$ loss. Peptide mass fingerprinting of the in-gel trypsin-digested MjGATase led to the identification of two unique peptides of masses 1,352.7 and 1,370.7 Da (Supplementary Fig. 2a). These masses are 17 Da lower and 1 Da greater than the expected mass of 1,369.7 Da for the peptide $V_{103}$YVDKE$N_{109}$DLFK$_{113}$ that has a single asparaginyl residue at position 109. Loss of $NH_3$ from N109 leading to succinimide formation would result in lowering of mass by 17 Da, while hydrolysis of the succinimide to aspartic acid (Asp) or isoaspartic acid (isoAsp) would lead to an increase in mass by 1 Da over that expected from the sequence of this peptide. Both peptides (1,352.7 and 1,370.7 Da) were subjected to fragmentation by matrix-assisted laser desorption ionization (MALDI), collision-induced dissociation (CID) and electron transfer dissociation (ETD)-MS/MS to validate the site of succinimide formation (Fig. 1c,d and Supplementary Fig. 2b–e). ETD-MS/MS of the 1,352.7 Da peptide showed $c_7 − 17$ and $z_5 − 17$ ion masses that are indicative of loss of $NH_3$ from N109 (Fig. 1c, inset). Further, MALDI and CID-MS/MS (Supplementary Fig. 2b,c) corroborated ETD-MS/MS data and indicated modification of N109 to succinimidyl moiety. MALDI and CID-MS/MS of 1,370.7 Da peptide suggest hydrolysis of succinimide as evident by an increase of 1 Da in the fragment ions ($b_7$ to $b_{10}$ and $y_5$ to $y_{10}$) masses over the expected masses (Supplementary Fig. 2d,e). This indicates the presence of Asp/isoAsp at position 109 in the 1,370.7 Da peptide. Fragmentation by CID-MS/MS that cleaves the bond between carbonyl carbon and NH of the peptide chain, (Fig. 2a) does not allow the distinction of Asp from isoAsp. However, fragmentation by ETD that cleaves between N–$C_\alpha$ and $C_\alpha$–$C_\beta$ bonds yields $c + 57$ and $z − 57$ ions that are diagnostic of isoAsp in the peptide[28–31] (Fig. 2b). The ETD-MS/MS of 1,370.7 Da peptide revealed the presence of isoAsp at position 109 (Fig. 1d), as evident from the presence of the diagnostic $c_6 + 57$ and $z_5 − 57$ fragment ions (Fig. 1d, inset). Detection of isoAsp in the peptide expected to contain asparaginyl residue at position 109 further supports modification of N109 to succinimide. It should be noted that unlike most of the reported cases where succinimide is rapidly hydrolysed to Asp/isoAsp[2,4,12,32], in MjGATase, a minor amount of the peptide containing the succinimide (abbreviated as Su in subsequent sections) was present even under conditions of in-gel trypsin digestion.

Translation of the DNA sequence of MjGATase yields a protein with a mass of 21,020 Da and an asparaginyl residue at position 109 (Supplementary Fig. 2f) confirming that the loss of 17 Da was indeed due to a post-translational modification. Further confirmation of succinimide formation at N109 was obtained by examining the mutants MjGATase_N109S and MjGATase_N109D. ESI-MS of MjGATase_N109S showed a molecular mass of 20,993 Da (Fig. 1e), in agreement with the theoretically predicted mass of 20,993 Da and indicated the

absence of the succinimide in this mutant. In contrast, ESI-MS of MjGATase_N109D showed the presence of two protein forms (Fig. 1f); a major population with a mass of 21,003 Da and a minor population having a mass of 21,021 Da (Fig. 1f, inset)

while the expected mass is 21,021 Da. The mass of 21,003 Da corresponds to the mass of MjGATase_N109D with succinimide and 21,021 Da corresponds to the mass of the native protein. Mass analysis of wild-type MjGATase and MjGATase_N109S in

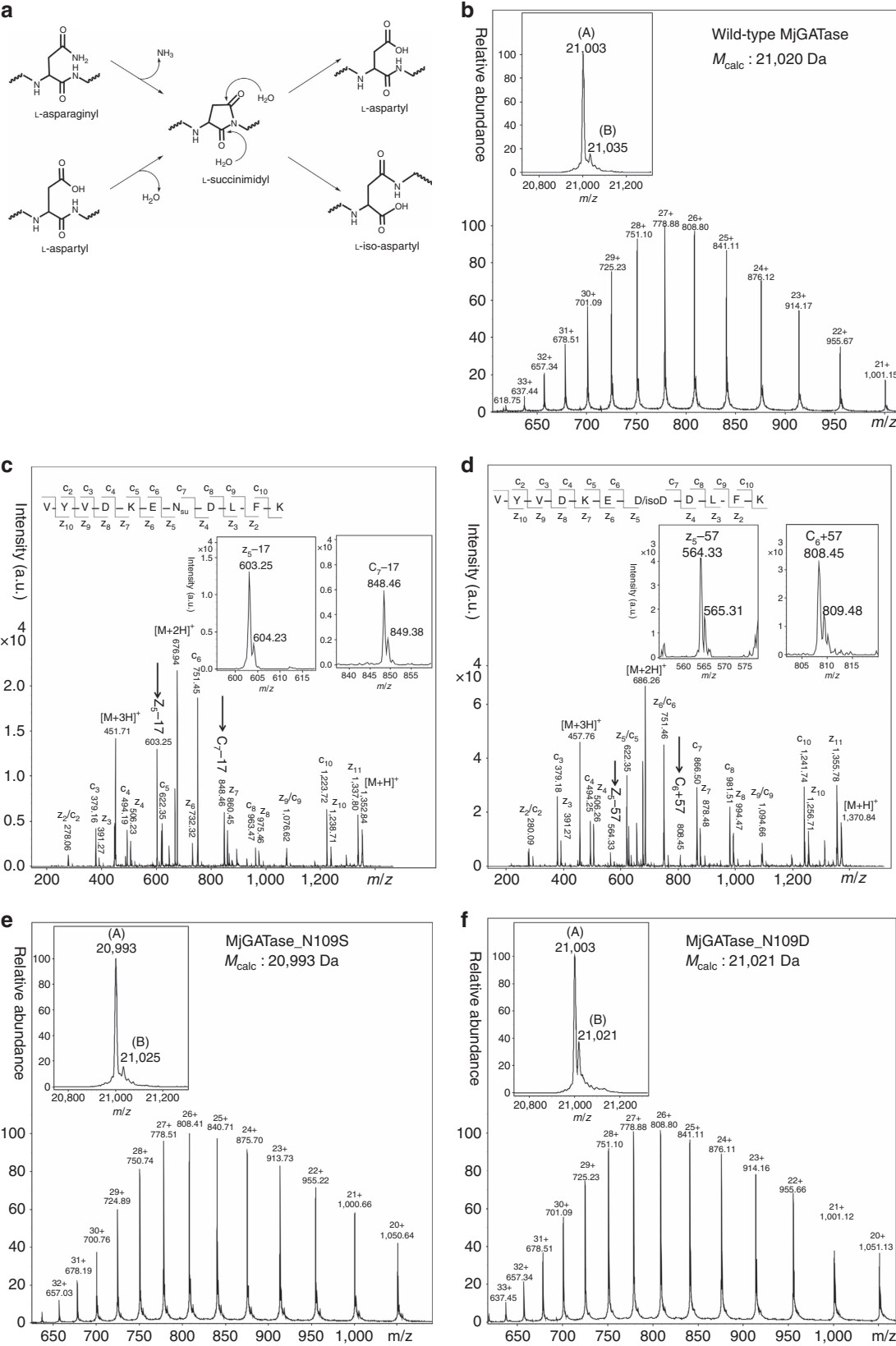

monoisotopic mode on an Orbitrap mass spectrometer also showed loss of 17 Da from the WT enzyme while the $M_{obs}$ of MjGATase_N109S was in complete agreement with the calculated monoisotopic mass of 20,979 Da (Supplementary Fig. 3). Taken together, the complete absence of succinimide in MjGATase_N109S and the formation of succinimide in MjGATase_N109D unequivocally establish that N109 is indeed the site of succinimide formation in MjGATase.

**The $n + 1$ residue (D110) stabilizes succinimide in MjGATase.** Among various factors that influence the susceptibility of Asn/Asp (residue n) to form succinimide, the following $n + 1$ residue is known to have the largest influence[33–36]. Hence, D110 was substituted with lysine (D110K) and glycine (D110G) to examine its functional role. ESI-MS of MjGATase_D110K yielded a molecular mass of 21,034 Da, in close agreement with the expected mass of 21,033 Da, indicating the absence of succinimide in this mutant (Fig. 3a). However, ESI-MS of MjGATase_D110G showed the presence of two species (Fig. 3b, inset); a major population with a mass of 20,963 Da, in close agreement with the expected mass of 20,962 Da and a minor species with a mass of 20,945 Da that is 17 Da lower than the predicted mass. The mass of 20,945 Da corresponds to the mass of MjGATase_D110G with N109 modified to succinimide, while the mass of 20,963 Da represents either the native protein (N109) or a protein species with the succinimidyl moiety hydrolysed to D109/isoD109. The absence of a stable succinimidyl moiety in MjGATase_D110K (Fig. 3a) and the presence of only a small proportion of protein with succinimide in MjGATase_D110G at pH 7.4 (Fig. 3b), suggests two possibilities; either succinimide formation is impaired or it is formed and subsequently, rapidly hydrolysed. Analysis of trypsin digested samples of the two mutants, MjGATase_D110K and MjGATase_D110G, showed the presence of two unique peptides of masses 995.5 Da and 1,312.7 Da, respectively (Supplementary Fig. 4a,b). These masses are 1 Da more than the expected masses of the peptide sequences VYVDKEN$_{109}$K$_{110}$(from MjGATase_D110K) and VYVDKEN$_{109}$G$_{110}$LFK (from MjGATase_D110G). An increase of 1 Da would be expected if N109 is first modified to succinimide and then hydrolysed to Asp/isoAsp. Indeed, MS/MS fragmentation of these two peptides by MALDI and CID showed the presence of aspartic acid at position 109 (Supplementary Fig. 4c–f). Further, ETD-MS/MS (Fig. 3c,d) showed the presence of isoAsp at position 109 as evident from the presence of diagnostic $c_6 + 57$ ion in VYVDKEN$_{109}$K$_{110}$ (Fig. 3c, inset) and $c_6 + 57$ and $z_5 - 57$ ions in VYVDKEN$_{109}$G$_{110}$LFK peptide fragments (Fig. 3d, inset). Detection of isoAsp at position 109 in the MjGATase_D110K and MjGATase_D110G confirms succinimide formation, followed by subsequent hydrolysis, suggesting the role of D110 in stabilizing the succinimide intermediate in MjGATase. Measurement of the mass of MjGATase_D110G over a range of pH established that the Asp/isoAsp in this variant reverts to succinimide which is

stably retained at low pH (Fig. 3e and Supplementary Fig. 5), while at neutral pH, the imide intermediate is rapidly hydrolysed (Fig. 3f and Supplementary Fig. 5). A similar analysis of MjGATase_D110K established succinimide formation at low pH (Supplementary Fig. 6). However, the level of succinimide in MjGATase_D110K was drastically lower than in MjGATase_D110G, indicating that the bulkier side chain of lysine impedes the cyclization to the imide intermediate. In conclusion, the regeneration of succinimide at low pH from Asp/isoAsp and its subsequent rapid hydrolysis, once reverted to neutral pH in MjGATase_D110G points to the importance of D110 in shielding the succinimide from hydrolysis by bulk water at neutral pH.

**Succinimide stabilizes the MjGATase structure.** Very similar far-ultraviolet and near-ultraviolet circular dichroism and tryptophan fluorescence spectra of wild type (WT) and mutants (MjGATase_N109S, N109D, D110K and D110G) of MjGATase at 25 °C (Supplementary Fig. 7) indicate that the succinimide formation does not significantly alter the overall structure of the enzyme at this temperature. A clear distinction in structural stability between the wild type, harbouring a succinimide and mutants, lacking the imide intermediate became evident from thermal and equilibrium unfolding studies. The WT$_{Su}$ and MjGATase_N109D$_{Su}$ enzymes do not unfold even at 100 °C. However, the mutants lacking succinimide start unfolding above 85 °C (Fig. 4a) and irreversibly precipitate at 100 °C. At 90 °C, the WT$_{Su}$ and MjGATase_N109D$_{Su}$ show no significant change in the circular dichroism (CD) spectrum while the mutants lacking the succinimide show complete loss of ellipticity (Fig. 4b). The relatively high $T_m$ of 85–90 °C for the mutants (MjGATase_N109S, D110G and D110K) that lack stable succinimide suggests that as in other thermophilic proteins the sequence of MjGATase confers a high degree of thermostability. However, the spontaneous formation of succinimide in the wild-type enzyme results in a marked enhancement of melting temperature, conferring hyperthermostabilty as evident by the absence of melting even at 100 °C.

Guanidinium chloride (GdmCl) induced unfolding, of WT and mutants of MjGATase, was followed by tryptophan fluorescence. The enzymes harbouring succinimide (WT$_{Su}$ and MjGATase_N109D$_{Su}$) remain well folded even in 8 M GdmCl, whereas mutants possessing no or little succinimide start unfolding above 4–5 M GdmCl (Fig. 4c). MjGATase has a single tryptophan at position 122 that emits at 322 nm in the absence of chaotrope. The red shift in the emission maximum to 344 and 347 nm in the fluorescence spectrum of MjGATase_N109S and MjGATase_D110K, respectively, indicates exposure of tryptophan to aqueous environment arising as a result of protein unfolding. A slightly smaller red shift to 338 nm is observed for MjGATase_D110G in 8 M GdmCl. These three mutants possessing no or little succinimide start unfolding above 4–5 M GdmCl. In contrast, largely similar emission spectra of WT$_{Su}$ and

**Figure 1 | N109 in *M. jannaschii* GATase is modified to succinimide.** (**a**) General mechanism for succinimide formation and hydrolysis. (**b**) LC-ESI-MS of WT MjGATase ($M_{calc}$ 21,020 Da). Inset: deconvoluted spectrum. Major species (A), $M_{obs}$ 21,003 Da and minor species (B), $M_{obs}$ 21,035 Da. The mass of 21,003 Da corresponds to WT$_{Su}$ while 21,035 Da corresponds to WT$_{Su}$ with the addition of two oxygen atoms. (**c**) ETD-MS/MS of 1,352.7 Da peptide showing loss of an $NH_3$ molecule from N109. Fragment ions ($z_5 - 17$ and $c_7 - 17$) corresponding to loss of 17 Da are shown in the inset and highlighted by the arrows. N$_{Su}$ represents asparaginyl residue that is deamidated to succinimide. (**d**) ETD-MS/MS of 1,370.7 Da peptide showing presence of isoAsp at position 109 arising from hydrolysis of succinimide. The inset shows $c_6 + 57$ and $z_5 - 57$ fragment ions, diagnostic of isoAsp. (**e**) LC-ESI-MS of MjGATase_N109S ($M_{calc}$ 20,993 Da). Inset: deconvoluted spectrum showing (A) of $M_{obs}$ 20,993 and (B) of $M_{obs}$ 21,025 Da. Mass of 20,993 Da corresponds to MjGATase_N109S, while 21,025 Da corresponds to the mass of MjGATase_N109S with the addition of two oxygen atoms. (**f**) LC-ESI-MS of MjGATase_N109D ($M_{calc}$ 21,021 Da). Inset: deconvoluted ESI spectrum showing presence of two forms; (A) ($M_{obs}$ 21,003 Da) and (B) ($M_{obs}$ 21,021 Da). Mass of 21,003 Da corresponds to N109D$_{Su}$, while 21,021 Da indicates the mass of native MjGATase_N109D.

**Figure 2 | Peptide backbone fragmentation by tandem mass spectrometry.** (**a**) Scheme showing fragment ion nomenclature and selected techniques for MS/MS fragmentation. Fragmentation by CID yields predominantly b and y ions arising from cleavage at the peptide bond while fragmentation by ETD results in cleavage of N and $C_\alpha$ bond giving rise to c and z ions. (**b**) Scheme showing fragmentation by electron transfer dissociation (ETD). The cleavage between N and $C_\alpha$ leads to the generation of c and z ions, while cleavage of the $c_\alpha$ and $c_\beta$ bond in isoAsp containing peptides gives rise to $c + 57$ and $z - 57$ ions.

MjGATase_N109D$_{Su}$ in the absence and in the presence of the chaotrope clearly indicate that the enzyme having succinimide resists unfolding by the chaotrope. The role of succinimide formation in imparting structural stability was also probed by near-ultraviolet CD. Near ultraviolet CD spectra recorded after prolonged incubation (12 h) of WT and mutants of MjGATase in high concentration (5 M) of GdmCl at 70 °C showed no significant change in the spectra of WT$_{Su}$ and MjGATase_N109D$_{Su}$. However, mutants (MjGATase_N109S, D110K and D110G) lacking succinimide showed a complete loss of near-ultraviolet CD band intensity, indicating loss of tertiary structure under similar conditions (Fig. 4d). Size-exclusion chromatography elution profiles of WT$_{Su}$ and MjGATase_N109S lacking the succinimide were identical in the absence of the GdmCl (elution volume of both proteins corresponds to a monomer mass of 21 kDa) (Supplementary Fig. 7d). This suggests that the presence or absence of succinimide does not affect the hydrodynamic radius of the two proteins. Chromatographic runs of the protein samples in the presence of 6 M GdmCl showed a large shift in the elution volume for MjGATase_N109S (Fig. 4e). On the contrary, WT$_{Su}$ showed only a marginal shift in elution volume (Fig. 4e). It should be noted that these protein samples were incubated in 8 M GdmCl solution at 37 °C for 12 h before size-exclusion chromatography. A large increase in the hydro-dynamic radius of MjGATase_N109S and only a marginal swelling of WT$_{Su}$ in the presence of high concentration of chaotrope clearly indicate that succinimide formation markedly enhances the stability of the overall fold of MjGATase, fully supporting the thermal unfolding results.

Activity measurements after preincubation of the protein samples at different temperatures showed that WT$_{Su}$ and MjGATase_N109D$_{Su}$ remain completely soluble and fully active up to 100 °C, whereas MjGATase_N109S, D110K and D110G start precipitating at temperatures above 85 °C. Reduction in the soluble form of MjGATase_N109S, D110G and D110K protein was reflected by a drastic drop in their enzymatic activity above 85 °C (Fig. 4f). Since MjGATase is completely inactive on its own and requires the presence of ligand bound *M. jannaschii* ATP pyrophosphatase (MjATPPase)[37], the effect of chaotrope on activity could not be measured as MjATPPase structure and subunit interaction were impaired in the presence of GdmCl.

Taken together, our experiments indicate that mutations (MjGATase_N109S, D110K and D110G) that do not enable side-chain-backbone cyclization destabilize the enzyme whereas a non-conserved mutation (N109D) that retains this ability is as stable as the wild-type protein. This indicates that the difference in the stabilities of the mutants lacking a stable succinimide and those (WT$_{Su}$ MjGATase and N109D$_{Su}$) harbouring the intermediate, is a result of side-chain-backbone cyclization and not due to the mutations per se.

**Remarkable stability of the succinimide in MjGATase**. The succinimide exhibited a remarkable stability in both WT$_{Su}$ and MjGATase_N109D$_{Su}$ enzymes. Despite incubation at 100 °C for 15 min, the two proteins did not exhibit any change in their molecular masses, confirming that succinimide is retained within the enzyme even after boiling (Fig. 5a, Supplementary Fig. 8a).

Incubation in 8 M GdmCl (12 h at 37 °C) revealed protein mass of 21,003 Da, corresponding to the protein with intact succinimide. The second species with a mass of 21,035 Da may be assigned to protein mass in which methionine is oxidized with retention of succinimide (Fig. 5b, Supplementary Fig. 8b).

Succinimide in MjGATase was also stable in a dilute solution of hydrochloric acid (0.1 N HCl) after incubation for 12 h at 37 °C (Supplementary Fig. 8c). Stability of this modification was also probed using $NH_2OH$, a stronger nucleophile than $H_2O$. The succinimide was found to be stable even after 2 h of

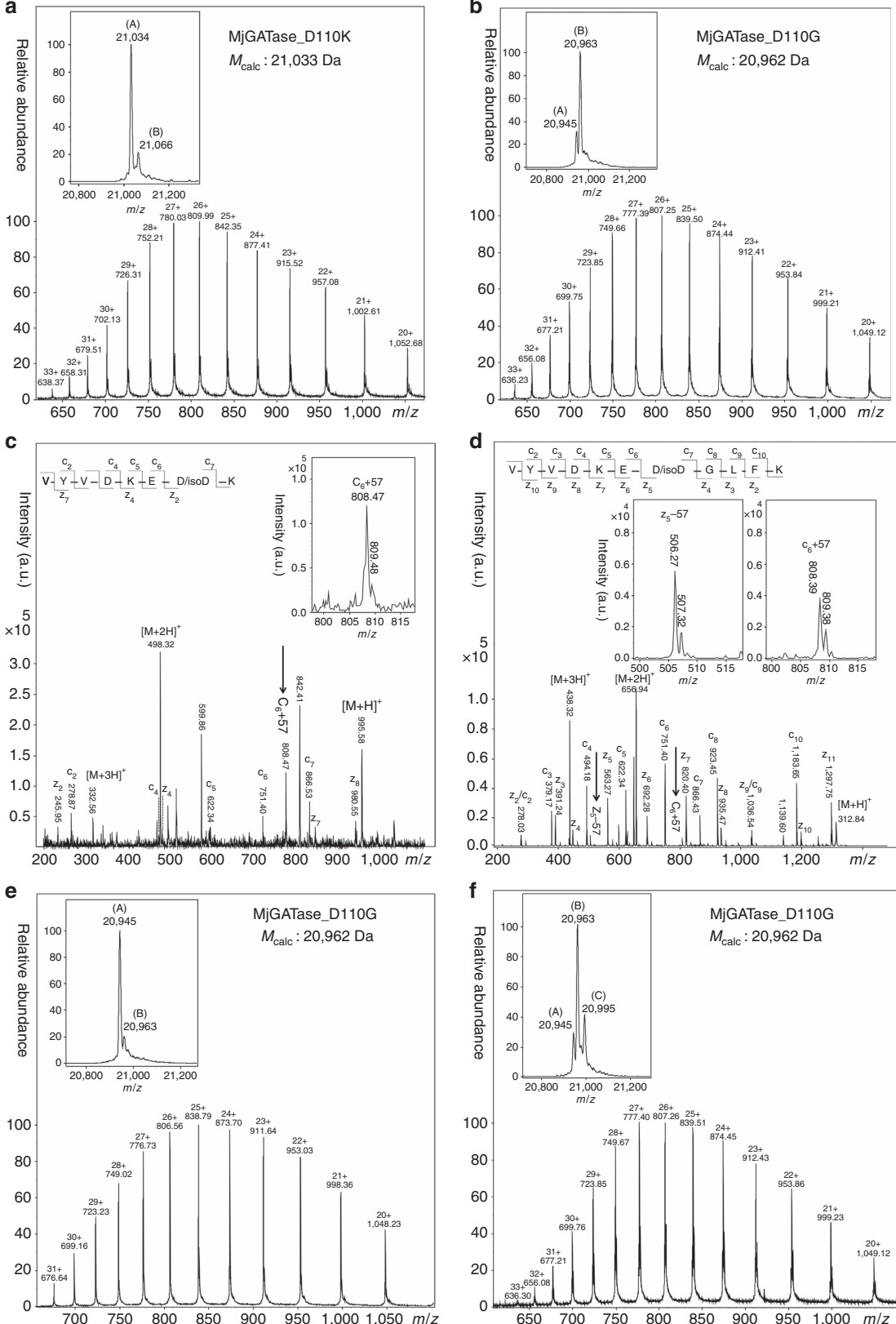

incubation in a solution containing 2 M $NH_2OH$ at 37 °C. Apart from the species with succinimide, the observed peaks of higher masses could correspond to either oxidized species or a hydroxymic acid derivative (Supplementary Fig. 8d). However, peptide chain cleavage as reported in earlier studies on peptides and proteins[38–40] was not observed in MjGATase, even after long exposure to $NH_2OH$.

**MjGATase and succinimide stabilities are interlinked.** The observations presented thus far suggest that the remarkable stability of MjGATase structure and the unusual stability of the succinimide are interlinked. Since MjGATase is resistant to unfolding at high temperature or in the presence of high concentrations of chaotrope, unfolding as a function of pH was examined to evaluate the role of protein structure on the stability of the succinimide. Fluorescence spectra acquired after incubation of MjGATase as a function of pH were suggestive of protein unfolding at high pH (Fig. 6a). Mass spectra recorded after incubation at extremes of pH (2.5 and 10.5) showed that the succinimide is stably retained within the enzyme at low pH, as supported by experimentally determined mass (21,003 Da) being lower by 17 Da over the theoretically predicted mass (21,020 Da) (Fig. 6b). However, mass spectra recorded after incubation at high pH (Fig. 6c) showed the presence of hydrolysed succinimide in two species: wild type as well as oxidized protein containing two oxygen atoms. This is evident from the experimentally determined masses, 21,021 and 21,053 Da being 1 and 33 Da more than theoretically predicted mass of 21,020 Da (Fig. 6c, inset). Similar results were obtained with MjGATase_N109D$_{Su}$ (Supplementary Fig. 9a,b). Electrophoresis under native conditions (native-PAGE) demonstrated increased mobility of the protein sample that was incubated at pH 10.5 over that of the samples incubated at pH 2.5 or 7.4 (Fig. 6d, Supplementary Fig. 12) suggesting hydrolysis of succinimide at high pH, yielding an additional negative charge (Asp/isoAsp) to the protein. Two-dimensional gel electrophoresis of MjGATase after preincubation at different pH also showed that high pH yields protein species with a pI that is distinct from that of the native sample (Supplementary Fig. 10). The lower pI value of the protein sample incubated at pH 10.5 indicates hydrolysis of succinimide to Asp/isoAsp.

The protein, unfolded at high pH reverts to the folded form with succinimide on lowering the pH albeit, with lower efficiency (Fig. 6e,f). The low efficiency of succinimide formation from Asp/isoAsp upon reversion to low pH could be attributed to the absence of precise fold required for spontaneous succinimide formation or due to a large accumulation of isoAsp in the hydrolysed sample. The near-complete conversion of the hydrolysed product to succinimide in MjGATase_D110G which was not unfolded by pre-exposure to high pH, lends support for

the dominant role of protein structure in enabling complete conversion to succinimide in MjGATase (Fig. 3e).

The above inference was further validated by examining the synthetic peptide, VYVDKENDLFK, for its ability to form and retain succinimide. Analysis of the synthetic peptide by MALDI-MS yielded a mass of 1,369.7 Da, in agreement with $M_{calc}$ of 1,369.7, showing the complete absence of succinimide (Supplementary Fig. 11a). Fragmentation of this peptide by MALDI and CID-MS/MS yielded the expected sequence (Supplementary Fig. 11b,c) with an asparaginyl residue. Whether the synthetic peptide forms succinimide which is subsequently, rapidly hydrolysed or its formation is impaired was probed by ETD-MS. Fragmentation by ETD-MS/MS showed the absence of the diagnostic $c + 57$ and $z − 57$ ions that would arise from an isoAsp containing peptide (Supplementary Fig. 11d). This shows that the absence of succinimide in the synthetic peptide is due to its inability to form succinimide and not due to the rapid hydrolysis of the intermediate. The complete absence of the succinimide intermediate in the synthetic peptide, in sharp contrast to the full-length MjGATase strongly supports the role of protein structure in enabling spontaneous succinimide formation and stabilization of the intermediate.

## Discussion

ESI-MS analysis of MjGATase, ETD-MS/MS of the tryptic peptide, $V_{103}$YVDKEN$_{109}$DLFK$_{113}$ and site-directed mutagenesis unequivocally establish that N109 is modified to succinimide in this enzyme. The presence of a single protein species in the mass spectrum of the freshly purified protein sample suggests the complete and rapid transformation of N109 to succinimidyl moiety. Although succinimide has been observed in some proteins such as recombinant human growth hormone, somatotropin, recombinant hirudin, lysozyme and certain monoclonal antibodies, this intermediate is evident only upon prolonged incubation at low pH or elevated temperature[41–47]. Further, in almost all cases reported, only a fraction of the total protein is found to possess the imide intermediate[41–45,47]. Rapid and complete transformation of asparaginyl/aspartyl residue to succinimidyl is not frequently observed in proteins under native conditions[3]. Succinimide formation requires the dihedral angles, $\psi$ and $\chi1$ to adopt the values $−120°$ and $+120°$ (Fig. 7a) respectively, that are energetically unfavourable and are generally forbidden in native well-folded proteins[3]. Also, the succinimide once formed, being sterically constrained would be prone to hydrolysis and hence short-lived.

The rate of deamidation is influenced by protein primary sequence, secondary, tertiary and quaternary structures, pH, temperature, ionic strength, dielectric constant and other solution properties[14,35,48,49]. Of these, the role of the $n + 1$ residue has been most comprehensively studied in proteins and

**Figure 3 | The $n + 1$ residue (D110) stabilizes succinimide in MjGATase.** (**a**) LC-ESI-MS of MjGATase_D110K ($M_{calc}$ 21,033 Da). Inset: deconvoluted spectrum shows two forms, (A) ($M_{obs}$ 21,034 Da) and (B) ($M_{obs}$ 21,066 Da). Species (A) corresponds to protein with hydrolysed succinimide while species (B) has two additional oxygen atoms. (**b**) LC-ESI-MS of MjGATase_D110G ($M_{calc}$ 20,962 Da). Inset: deconvoluted spectrum shows two species; (A) ($M_{obs}$ 20,945 Da) and (B) ($M_{obs}$ 20,963 Da). Mass of 20,945 Da corresponds to MjGATase_D110G with succinimide (MjGATase_D110G N109$_{Su}$) and 20,963 Da represents MjGATase_D110G with hydrolysed succinimide (MjGATase_D110G N109**$_{D/isoD}$**). ETD-MS/MS of (**c**) 995.5 Da and (**d**) 1,312.7 Da peptides derived from in-gel trypsin digestion of MjGATase_D110K and D110G, respectively. The spectra show presence of isoAsp at position109 in both mutants. Insets show $c_6 + 57$ and, $c_6 + 57$ and $z_5 − 57$ fragment ions in the peptides from MjGATase_D110K and MjGATase_D110G, respectively. (**e**) LC-ESI-MS of MjGATase_D110G ($M_{calc}$ 20,962 Da) recoded after incubation for 12 h at pH 2.5. Inset: deconvoluted spectrum indicates the presence of two species; (A), $M_{obs}$ 20,945 Da and (B), $M_{obs}$ 20,963 Da. Species (A) is MjGATase_D110G N109$_{Su}$ and species (B) is MjGATase_D110G N109**$_{D/isoD}$**. (**f**) LC-ESI-MS of MjGATase_D110G ($M_{calc}$ 20,962 Da) recorded after reverting the pH to 7.4 from 2.5. It should be noted that this protein sample was initially kept in a solution of pH 2.5 for 12 h followed by further incubation for 12 h at pH 7.4. Inset: deconvoluted spectrum shows presence of three species, (A) ($M_{obs}$ 20,945 Da), (B) ($M_{obs}$ 20,963 Da) and (C) ($M_{obs}$ 20,995 Da). Mass of 20,945 Da corresponds to MjGATase_D110G N109$_{Su}$, 20,963 Da represents MjGATase_D110G N109**$_{D/isoD}$** and 20,995 Da corresponds to MjGATase_D110G with hydrolysed succinimide and methionine oxidized at two sites.

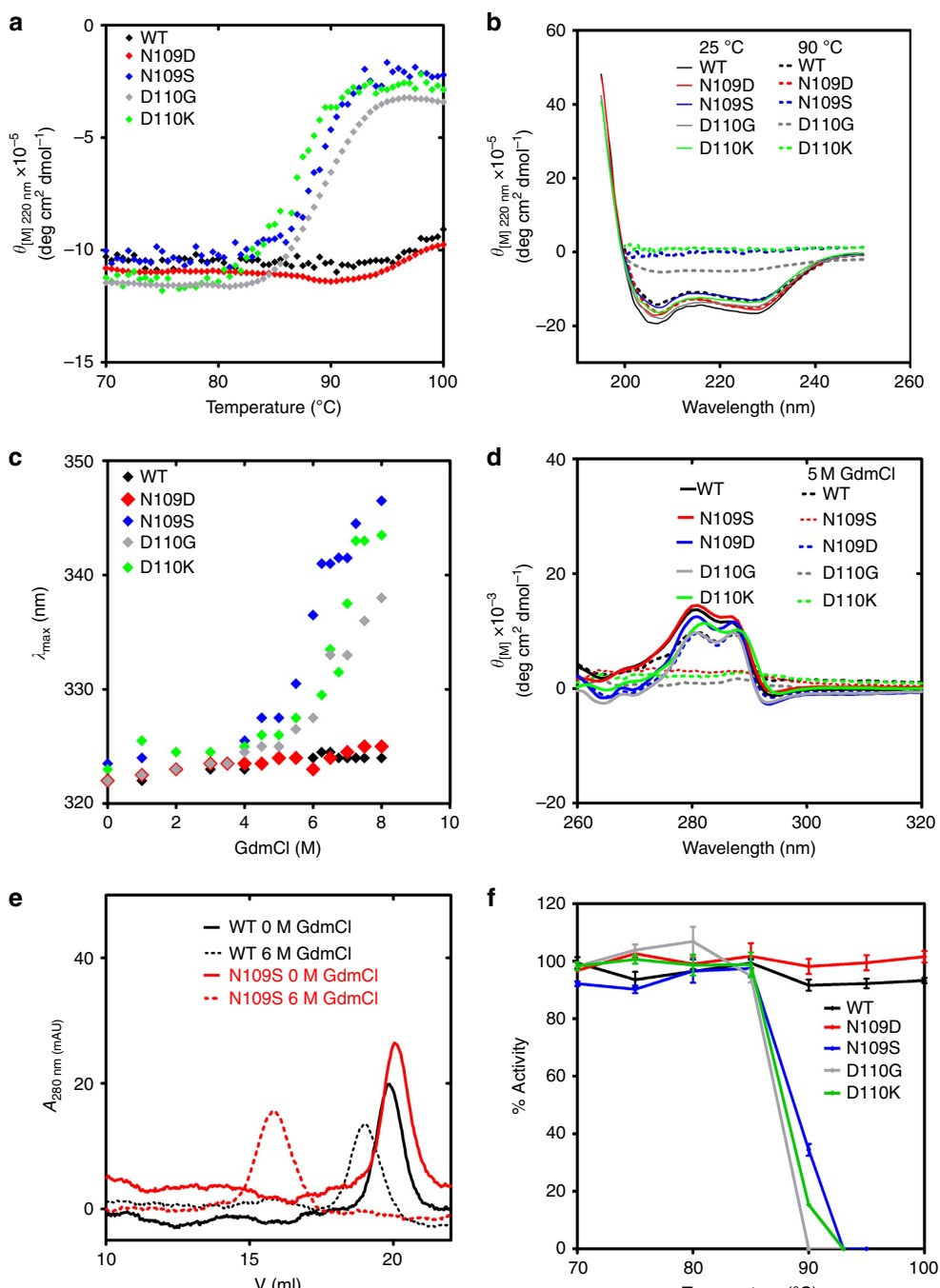

**Figure 4 | Succinimide confers structural stability to MjGATase.** (**a**) Molar ellipticity recorded at 220 nm using a CD spectropolarimeter as a function of temperature with a ramp rate of 0.5 °C min$^{-1}$. (**b**) Far-ultraviolet CD spectra of WT and mutants of MjGATase recorded after incubation at 25 and 90 °C for 30 min. (**c**) Fluorescence emission maximum of WT and mutants of MjGATase as a function of GdmCl concentration. Fluorescence spectra were recorded after incubation of protein samples in solution with different concentrations of GdmCl at 37 °C for 12 h. (**d**) Near-ultraviolet CD spectra of WT and mutants of MjGATase recorded after incubation of protein samples in 5 M GdmCl containing solution at 70 °C for 12 h. (**e**) Size-exclusion chromatography elution profile of WT$_{Su}$ (enzyme harbouring the succinimide) and N109S mutant (enzyme lacking succinimide) in the absence and presence of GdmCl. (**f**) Enzymatic activity of the soluble fraction after preincubation at different temperatures for 30 min. The time period for incubation at 100 °C was 15 min. The levels of soluble protein in the case WT$_{Su}$ and N109D$_{Su}$ mutant did not change over the control while the protein amounts in MjGATase_N109S, D110K and D110G after heating above 85 °C, were below the detection limit of Bradford assay.

synthetic peptides[3,33–36]. Protein sequences that undergo asparaginyl deamidation/aspartyl isomerization show a high prevalence of glycine at the $n+1$ position[34,35,50]. Glycine lacks side chain and hence, does not impose steric hindrance for the nucleophilic attack of its NH on the carbonyl carbon of Asn side chain, required for succinimide formation. Another unique feature of glycine is its large conformational flexibility. Other than steric factors, various other mechanistic roles such as, stabilization of the backbone nitrogen anion for nucleophilic attack on the carbonyl carbon of the side chain of Asp/Asn, deprotonation of peptide NH to form the succinimide and protonation of leaving group (NH$_2$/OH of Asn/Asp) are

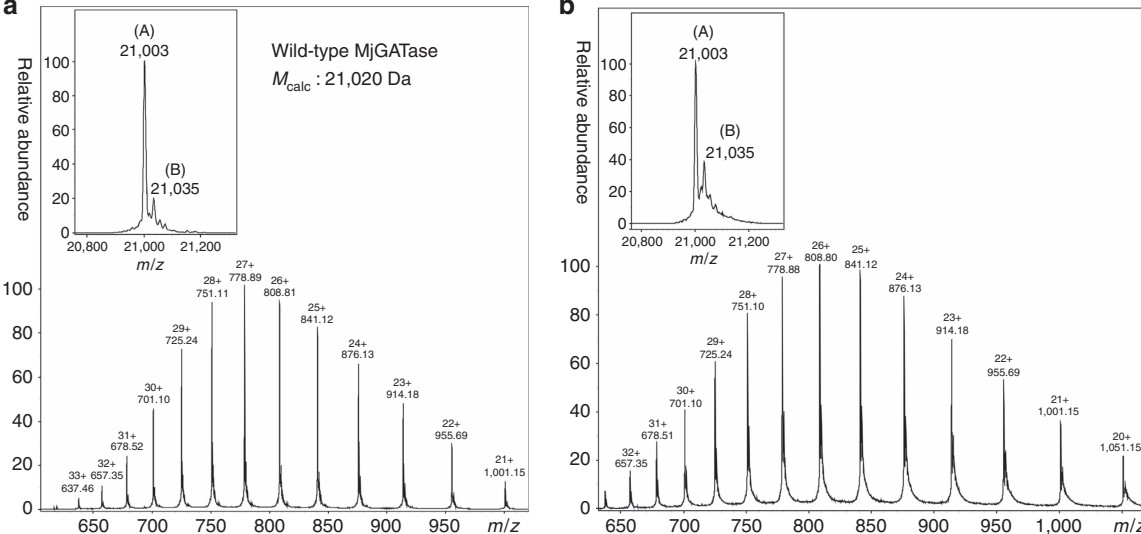

**Figure 5 | Succinimide in MjGATase is remarkably stable. (a)** LC-ESI-MS of WT MjGATase ($M_{calc}$ 21,020 Da) recorded after incubation at 100 °C for 15 min. Inset shows deconvoluted spectrum with two species; (A) ($M_{obs}$ 21,003 Da) and (B) ($M_{obs}$ 21,035 Da). **(b)** LC-ESI-MS of WT MjGATase recorded after incubation in 8 M GdmCl for 12 h at 37 °C. The deconvoluted spectrum in the inset shows the presence of two protein species; (A) ($M_{obs}$ 21,003 Da) and (B) ($M_{obs}$ 21,035 Da). Mass of 21,003 Da corresponds to WT$_{Su}$, while 21,035 Da corresponds to WT$_{Su}$ with the addition of two oxygen atoms as a result of methionine oxidation.

attributed to the neighboring $n + 1$ residue. However, thus far, to the best of our knowledge, there is no direct evidence for the role of $n + 1$ residue in imparting stability to the succinimide. Our studies provide the first direct evidence for the role of the $n + 1$ residue in conferring stability to the succinimide. The MS analysis of MjGATase_D110K and D110G show that though these mutants can form succinimide, a mutation at the $n + 1$ residue leads to rapid hydrolysis of the intermediate; a feature, unlike the WT$_{Su}$ enzyme. Detection of isoAsp in the peptides of D110 mutants, formation of succinimide from Asp/isoAsp at low pH in MjGATase_D110G and its rapid hydrolysis upon reversion to neutral pH clearly support the essentiality of D110 in preventing hydrolysis of succinimide. The structure of *Pyrococcus horikoshii* glutaminase (PhGATase, PDB id 1WL8), a homologue of MjGATase also shows the presence of succinimide where the residue involved in cyclization is D112. The dihedral angles, $\psi$ and $\chi 1$ of succinimide at position 112 in PhGATase are $-122°$ and 120°, respectively, suggesting that these rotamer states should have been significantly populated, thereby facilitating cyclization. These conformational requirements must have also been met for N109 in MjGATase, as this residue is completely modified to the succinimidyl form. Further, the crystal structure of PhGATase shows that the side chain of the $n + 1$ residue, E113 has flipped over the succinimide and appears to contribute to the microenvironment that shields the succinimide from nucleophilic attack by bulk water. The side chain and backbone atoms of the residues D110, E111, K116, L142 and Y161 that surround and bury the succinimide (Fig. 7b) are largely conserved in MjGATase. It is interesting to note that though three water molecules are present at 4 Å distance cutoff (Fig. 7b), the imide intermediate remains unhydrolysed.

Two recent examples of the succinimidyl group in the crystal structure of proteins include *Thermus thermophilus* amylomaltase and *Ficus benghalensis* peroxidase but neither the spontaneity of formation, nor the functional significance of this modification has been examined[51,52]. The succinimide in MjGATase is fully retained even after long periods of storage, boiling and prolonged incubation in high concentrations of chaotrope, strongly suggesting that this modification is remarkably stable in this

enzyme. The comparative analyses of thermal and equilibrium unfolding studies of MjGATase with succinimidyl (WT$_{Su}$) and mutants lacking succinimidyl moiety establish that this post-translational modification aids in maintaining the structural stability of the enzyme at high but physiologically relevant temperatures with regard to the habitat of this hyperthermophilic archaean. These observations are novel and are counterintuitive to the existing paradigm where succinimide-mediated changes are widely believed to be one of the potential causes for loss of protein structure due to the accumulation of D-Asp and D/L-isoAsp[11,21,53]. High activity of the repair enzyme, protein isoaspartyl methyltransferase (PIMT) that repairs isoAsp to Asp and significant reduction in the number of asparagine and glutamine residues in the protein sequences of thermophiles as compared with their mesophilic homologs provide some indirect evidence in support of the existing paradigm that invokes a largely detrimental effect of succinimide on protein structure and function[21,54,55]. Our findings on the remarkably stable succinimide in MjGATase strongly suggest the evolutionary selection of a sequence and thereby, a structure that favours spontaneous succinimide formation and protects the intermediate from hydrolysis even under extreme conditions. Further, unfolding of MjGATase being concomitant with hydrolysis of succinimide at high pH implicates the role of three-dimensional structure in endowing remarkable stability to the succinimide. We propose that protein and succinimide stabilities in MjGATase are interlinked with the protective protein scaffold preventing hydrolysis of succinimide and the succinimide, in turn, restricting the conformational dynamics of the loop harbouring this modification through constraining the dihedral angle $\psi$. In the structure of PhGATase, the loop containing succinimide connects two $\beta$ strands that bear the residues Y101 and W125, which contact each other and also the loop harbouring the catalytic residues His166 and Glu168 (Fig. 7c). The succinimide modification probably enables precise positioning of these residues by constraining loop dynamics. The activity of MjGATase being conditional to the binding of ligand complexed MjATPPase, is tightly regulated. Restricted loop dynamics through succinimide could confer adaptive

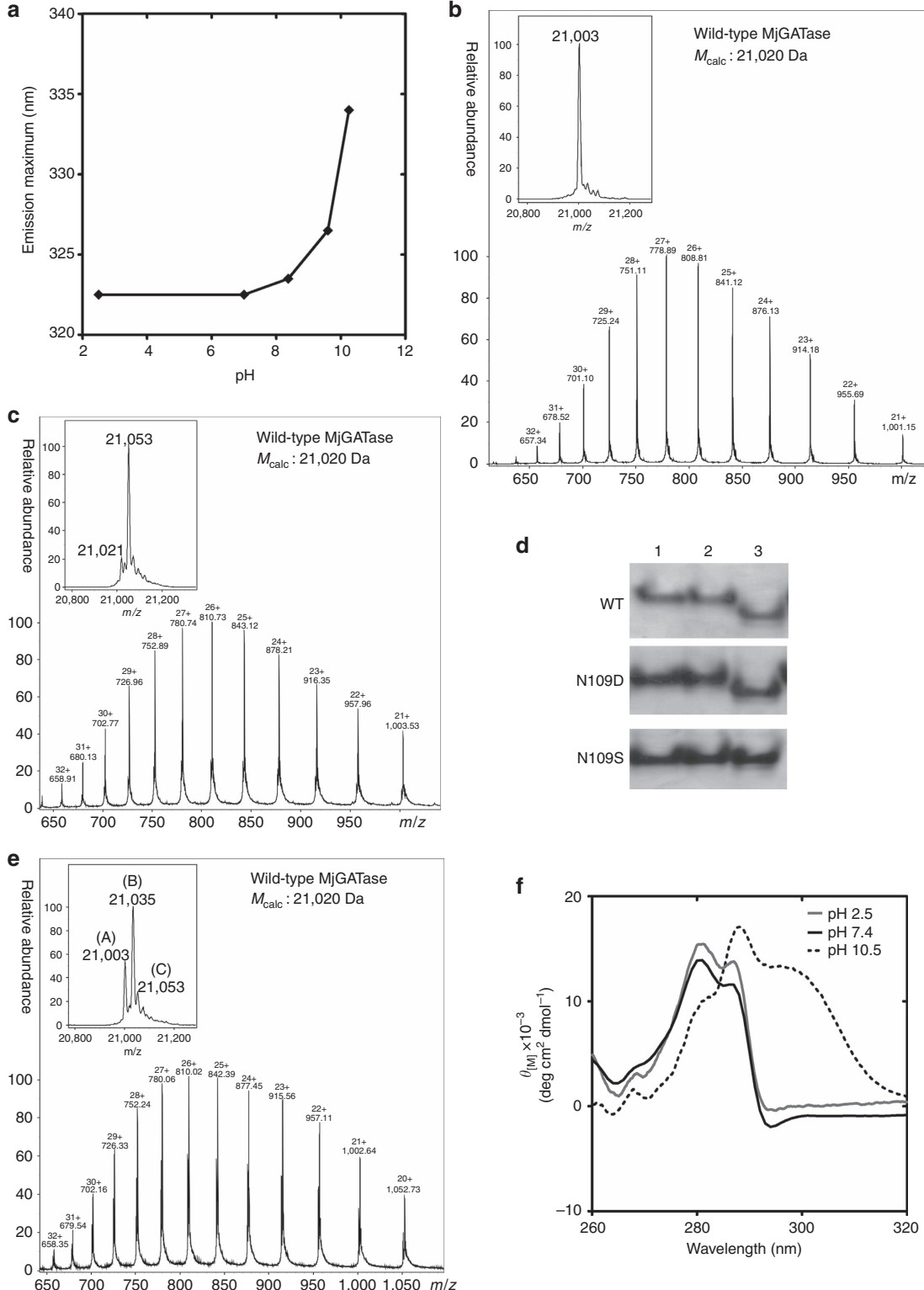

**Figure 6 | MjGATase and succinimide stabilities are inter-linked.** (**a**) Fluorescence emission maximum versus pH. LC-ESI-MS of WT MjGATase ($M_{calc}$ 21,020 Da) recorded after incubation at (**b**) pH 2.5 and (**c**) pH 10.5 for 12 h at 37 °C. The deconvoluted spectrum in the inset to **b** shows a $M_{obs}$ of 21,003 Da corresponding to the mass of $WT_{Su}$. Inset to **c** shows (A) of $M_{obs}$ 21,021 Da that corresponds to WT with hydrolysed succinimide (Asp/isoAsp) and (B) of $M_{obs}$ 21,053 Da is WT with hydrolysed succinimide and two additional oxygen atoms. (**d**) Native-PAGE of WT, MjGATase_N109D and MjGATase_N109S. The lanes 1, 2 and 3 correspond to pH 2.5, 7.4 and 10.5, respectively of the incubation buffers. Full gel picture is shown in Supplementary Fig. 12. (**e**) LC-ESI-MS of $WT_{Su}$ enzyme sample, initially kept in a solution of pH 10.5 for 12 h followed by further incubation for another 12 h at pH 2.5. The deconvoluted spectrum in the inset shows presence of three species; (A) ($M_{obs}$ 21,003 Da) is $WT_{Su}$, (B) ($M_{obs}$ 21,035 Da) is $WT_{Su}$ with addition of two oxygen atoms and (C) ($M_{obs}$ 21,053 Da) is WT (D/isoD) along with two additional oxygen atoms. (**f**) Near-ultraviolet CD spectrum of WT MjGATase showing refolding of the protein sample after reversion of pH to 2.5 and 7.4 from 10.5.

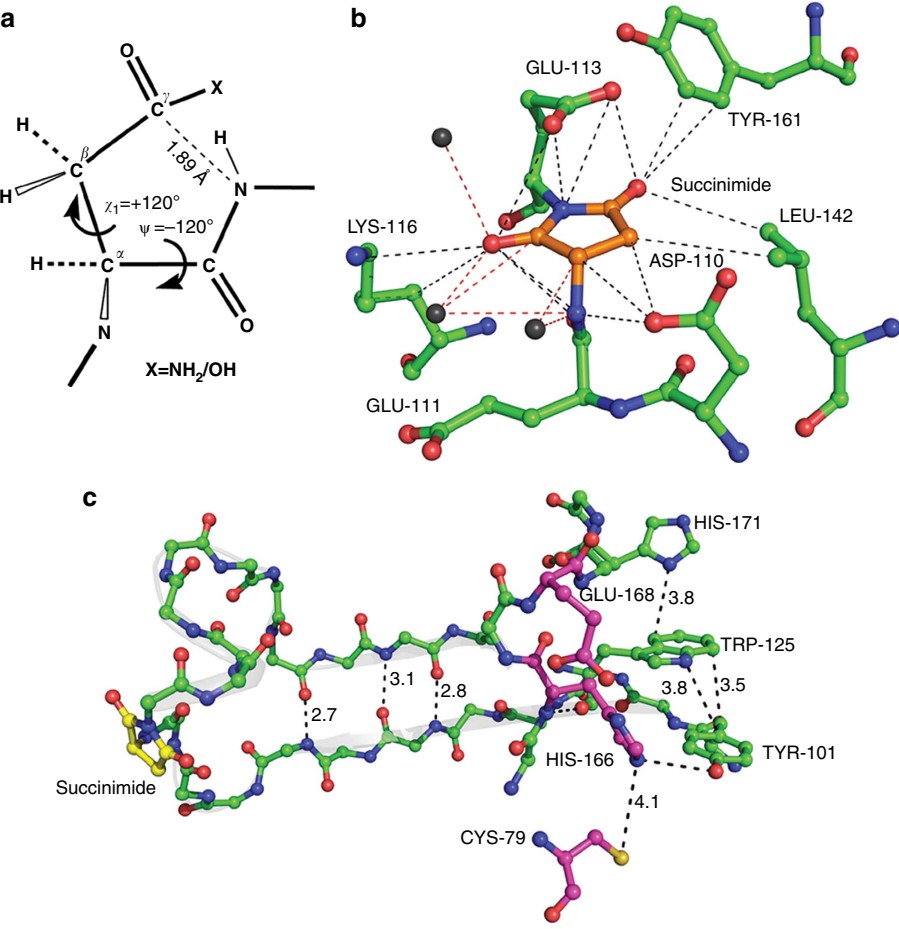

**Figure 7 | Succinimide imparts rigidity to peptide backbone.** (**a**) Productive conformation for succinimide formation. This conformation ($\psi = -120°$ and $\chi1 = +120°$) is generally forbidden in native well-folded protein structures. (**b**) Residues in contact with succinimide within 4 Å cutoff in PhGATase structure (PDB id: 1WL8). Water molecules are indicated by non-bonded black spheres. (**c**) A portion of PhGATase structure showing two β-strands connected by a loop harbouring succinimide at position 112. Succinimide (highlighted in yellow), connects the two β strands that harbour Y101 and W125, that in turn contact active site residues (highlighted in magenta).

advantage to *M. jannaschii* GATase, as it may enable the enzyme to resist high thermal fluctuations and retain tightly regulated activity at high temperatures. We suggest that succinimide mediated rigidity could be a novel mechanism for enhanced structural stability in hyperthermophiles.

## Methods

The primers custom synthesized at Sigma-Aldrich, India. Phusion high-fidelity DNA polymerase and dNTPS were procured from Thermo Scientific, USA. All clones were sequenced at the in-house genomics facility. Media components for growing *Escherichia coli* cells were from HiMedia, Mumbai, India. Superdex 200 and Q-sepharose resins for chromatography were from GE Health Care Life Sciences, UK. Matrices for MALDI-TOF mass spectrometry were either from Bruker Daltonics, Germany, or Sigma-Aldrich, USA. All chemical reagents for enzyme assay were of the highest quality and purchased from Sigma-Aldrich, USA. Organic solvents for mass spectrometry and reverse-phase high-performance liquid chromatography (RP-HPLC) were of highest purity, procured from Sigma-Aldrich, USA or Merck, Germany.

**Site-directed mutagenesis.** The MjGATase mutants were generated by the quick-change PCR method following manufacturer's protocol (Invitrogen Life Technologies, USA). Wild-type MjGATase clone (pST39_WTMjGATase) was used as a template for PCR. The primers used to introduce the mutations are listed in Supplementary Table 1. Introduction of the desired mutation was confirmed by DNA sequencing.

**Protein expression and purification.** MjGATase gene was cloned into the pET3atr vector using the restriction enzyme sites NdeI and BamHI to yield the plasmid

pET3atr_MjGATase. Subsequently, MjGATase gene fragment was excised out from pET3atr_MjGATase using the restriction enzymes XbaI and BamHI and sub-cloned into cassette I of pST39 polycistronic expression vector to generate pST39_MjGA-Tase[56]. The sequence of the gene was confirmed by DNA sequencing and was found to be identical to that deposited in the NCBI database (gene id: 1,452,484). To hyper-express MjGATase, Rosetta (DE3) pLys S *E. coli* cells carrying pST39_MjGATase plasmid were grown at 37 °C to an OD of 0.4–0.6, induced with 0.3 mM of IPTG and grown further for another 6–8 h at 37 °C. Cells were collected by centrifugation at 4,000*g*, resuspended in a buffer containing 20 mM Tris HCl, pH 7.4, 10% glycerol, 2 mM DTT, 0.1 mM PMSF, and 0.1 mM EDTA and lysed using a French press. The cell lysate was centrifuged at 30,500*g* for 30 min to remove the cell debris. Supernatant was heated at 70 °C for 30 min (this step precipitates bacterial proteins) and centrifuged at 30,500*g* for 30 min. Polyethyleneimine (0.01%) was added to the supernatant to precipitate nucleic acids and centrifuged at 30,500*g* for 30 min. Thereafter, the supernatant was loaded on to a Q-sepharose column (HR 16/10) attached to AKTA Basic HPLC (GE Healthcare) for anion exchange chromatography. The protein was eluted using a linear gradient of NaCl in buffer A (20 mM Tris HCl, pH 8.0, 10% glycerol, 2 mM DTT, 0.1 mM PMSF and 0.1 mM EDTA).The fractions containing pure protein were pooled and dialysed against buffer A. The protein was concentrated using Amicon ultra protein concentrator (Millipore) with 10 kDa cutoff. The purified protein was analysed by SDS–polyacrylamide gel electrophoresis (PAGE)[57]. The protein concentration was estimated by the method of Bradford using BSA as standard[58].

(His)₆-tagged MjGATase was cloned, expressed and purified as described below. Gene for MjGATase was PCR amplified using pST39_WTMjGATase plasmid as a template and cloned into modified pET21b vector using BamHI and SacI restriction sites to generate protein with N-terminal (His)₆-tag. The sequence of forward and reverse primers used for PCR amplification is provided in Supplementary Table 1. The clone was confirmed by sequencing. This clone yields protein containing five extra residues, three residues (Met, Ala and Ser) before and

two (Gly and Ser) after the (His)$_6$-tag. These residues are derived from the vector sequence and the BamHI restriction site. Conditions for cell growth and induction of protein expression were similar to that described for untagged MjGATase. Cells were harvested by centrifugation at 4,000$g$, resuspended in lysis buffer (20 mM Tris HCl, pH 7.4, 10% glycerol, 2 mM DTT, 0.1 mM PMSF and 0.1 mM EDTA) and lysed using a French press. The cell lysate was centrifuged at 30,500$g$ for 30 min to remove the cell debris. The supernatant was mixed with Ni-NTA agarose beads (Thermo Fisher Scientific, USA) and allowed to bind for 4 h, followed by four washes. The composition of wash buffers was, wash I, lysis buffer; wash II, lysis buffer containing 10 mM imidazole; wash III, lysis buffer containing 25 mM imidazole and wash IV, lysis buffer containing 50 mM imidazole. The recombinant (His)$_6$-tagged protein was eluted using lysis buffer containing 250 mM imidazole. The fractions containing pure protein, as determined by SDS–PAGE were pooled, concentrated using Amicon ultra protein concentrator with 10 kDa cutoff (Millipore) and further purified by size-exclusion chromatography using HiLoad 16 × 600 mm Superdex 200 (preparatory grade) column connected to AKTA Basic HPLC (GE Healthcare) with buffer A (20 mM Tris HCl, pH 7.4, 10% glycerol, 2 mM DTT, 0.1 mM PMSF) as the mobile phase.

**Mass spectrometric analyses of intact proteins and peptides.** Mass spectrometric measurements of all proteins and their tryptic digest were done at the mass spectrometry facility at the Molecular Biophysics Unit, Indian Institute of Science, Bangalore, India. In addition, the ESI-MS of WT MjGATase and MjGATase_N109S were recorded on a Q Exactive hybrid quadrupole-Orbitrap mass spectrometer (Thermo Fisher Scientific, USA) at Department of Chemistry, Physical and Theoretical Chemistry Laboratory, University of Oxford.

Molecular masses of intact proteins were obtained using an ESI-Q-TOF mass spectrometer (Maxis Impact, Bruker Daltonics, Germany), equipped with an HPLC system (Agilent Technologies, USA). Source conditions for spectral acquisition on ESI-Q-TOF included dry heater temperature of 200 °C, nebulizer pressure of 1.8 bar, dry gas flow rate of 9.0 l min$^{-1}$, capillary voltage of 4,500 V, endplate offset voltage of 500 V and charging voltage of 2,000 V. Protein sample (600 fmol to 2 pmol) was injected on to a reverse-phase C8 column (3 × 100 mm, 3.5 µm particle size, Agilent Technologies, USA). Samples were eluted using water, acetonitrile mix with 0.1% formic acid as the mobile phase with a flow rate of 0.2 ml min$^{-1}$. The Agilent tuning mix which included ions with $m/z$ in the range of 118.08 to 2,721.89 Da was used for calibration. Data were analysed using the Bruker Daltonics Data analysis 4.1 software. LC-ESI-MS was also used to assess the stability of succinimide within MjGATase after exposure to high temperature (100 °C), high concentration of chaotrope (8 M GdmCl), extremes of pH (2.5 and 10.5), 0.1 N HCl and 2 M NH$_2$OH. For this 10 µM MjGATase was preincubated at 100 °C for 15 min, in 8 M GdmCl, in 0.1 N HCl, and in 50 mM glycine at pH 2.5 and 10.5 for 12 h at 37 °C before MS analysis. Incubation in 2 M NH$_2$OH was for 2 h at 37 °C. The level of succinimide as a function of pH in MjGATase_D110G and D110K was analysed by ESI-MS after preincubation for 6 h in solutions of 50 mM glycine at different pH.

In addition, the masses of WT MjGATase and MjGATase_N109S were also obtained using ESI-MS on Orbitrap mass spectrometer. The protein samples were dissolved in water and then zip-tipped to 1:1 mixture of H$_2$O-acetonitrile with 0.1% formic acid using a C4 column. The spectra were acquired using direct injection in the off-line nano-ESI mode using a silica capillary, at 70,000 resolution with 15 micron scans and averaged with a noise level parameter of 4.68. Data were viewed using Xcalibur 2.2 SP1.48 (Thermo Fisher Scientific). For isotopically resolved data, the Xtract algorithm, licensed as part of Qual Browser in Xcalibur 2.2 SP1.48, was used for peak picking and deconvolution.

In-gel trypsin digestion was performed following the protocol described by Shevchenko et al.[59], but with minor modifications to enable detection of succinimide. An amount of 25–30 µg of purified MjGATase was subjected to SDS–PAGE and the gel was stained with Coomassie Brilliant Blue R-250 to visualize the protein band. Protein bands on the gel were excised out, cut into small pieces (1 mm$^3$) and washed twice with water followed by destaining using a 1:1 mixture of methanol and 50 mM Tris HCl, pH 7.0. The gel pieces were dehydrated for 5 min using first, a 1:1 mixture of acetonitrile and 50 mM Tris HCl, pH 7.0 followed by 100% acetonitrile for 30 s. The gel pieces were dried in a speed-vac for 10 min, rehydrated by incubating at 56 °C for 20 min in a buffer containing 25 mM dithiothreitol and 50 mM Tris HCl, pH 7.0. Thereafter, the gel pieces were washed twice with water, dehydrated and dried as before. The dried gel pieces were rehydrated in 50 mM Tris HCl, pH 7.0 containing 350 ng of sequencing-grade trypsin (Promega, USA) and incubated overnight at 37 °C. It should be noted that all the steps of in-gel trypsin digestion were performed at pH 7.0 to avoid hydrolysis of succinimide or deamidation that may occur at pH 8.0 (ref. 60), a condition routinely used for trypsin digestion. The amount of trypsin was doubled to overcome its lower activity at pH 7.0. Further, alkylation by iodoacetamide was also omitted. The solution containing the digested peptides was transferred into a fresh tube, 50% acetonitrile containing 5% formic acid was added to the gel pieces, vortexed for 20 min and sonicated for 15 min to extract the peptides entrapped in the gel pieces. This step was repeated once again, all the fractions were pooled and the volume was reduced to 10–20 µl on a speed-vac. The tryptic peptide fragments were analysed by MALDI, CID and ETD mass spectrometry. For MS analysis of MjGATase peptides by MALDI, 1 µl of trypsin digested MjGATase was mixed with

equal volume of saturated solution of α-cyano-4 hydroxycinnamic acid prepared a in 1:1 mixture of H$_2$O-acetonitrile with 0.1% triflouroacetic acid, spotted on to the target plate and air dried. The data were acquired using the Flex Control software and were analysed using Flex analysis software version 3.4. The MALDI-TOF instrument was calibrated in the positive ion mode using a mixture of peptides with $m/z$ in the range of 757–3,147 Da before data acquisition. Mass spectrometric analyses, by CID and ETD, on the peptides, were carried out on an ion-trap mass spectrometer (HCT Ultra ETD II, Bruker Daltonics, Germany) equipped with RP-HPLC (Aligents Technologies, USA). The data were acquired in auto MS/MS mode with both CID and ETD fragmentation methods. The instrument was calibrated in the positive ion mode using Agilent tuning mix that included $m/z$ in the range of 118.08–2,721.89 Da. Data were analysed using the Bruker Daltonics Data analysis 4.1 software.

Fmoc solid-phase peptide synthesis protocol was used to synthesize an 11-residue peptide corresponding to the tryptic fragment V$_{103}$YVDKENDLFK$_{113}$ of MjGATase. A PS3 peptide synthesizer (Protein Technologies, Arizona, USA), on a target scale of 0.1 mol equivalent, corresponding to 300 mg of resin was used to carry out the synthesis. The tertiary butyl group was used to protect the side chains of Asp and Glu while Asn and Lys side chains were protected by a trityl group and tert-butoxy carbonyl group, respectively. After the synthesis, the peptide was cleaved off the resin and deprotected using reagent K (TFA/phenol/anisole/water/EDTA (82.5:5:5:5:2.5)). Thereafter, the resin was filtered off and trifluoroacetic acid (TFA) was removed by vacuum centrifugation. The peptide was precipitated with diethyl ether. The precipitate was washed several times with ether, centrifuged and the crude peptide isolated by decanting the supernatant. The crude peptide was purified by RP-HPLC using a C18 column (Phenomenex, 9.4 mm × 250 mm, 5–10 µm particle size). Acetonitrile/H$_2$O/0.1% TFA solvent system was used as the mobile phase. The peptide was eluted using a linear gradient of 3–100% acetonitrile over 25 min with a flow rate of 1.0 ml min$^{-1}$. The peptide fractions were detected at 226 nm. The purified synthetic peptide was examined for its ability to form succinimide by mass spectrometry.

**Native-PAGE.** A concentration of 50 µM of MjGATase was incubated in 50 mM glycine of pH 2.5, 7.4 and 10.5 at 37 °C for 12 h. Thereafter, the pH of all the protein samples was adjusted to 7.4 before their analysis by native-PAGE. Protocol for native-PAGE was essentially similar to SDS–PAGE[57] except for the omission of SDS, β-mercaptoethanol and boiling.

**Two-dimensional gel electrophoresis.** A concentration of 100 µM of MjGATase was incubated in 100 mM of Tris-glycine, pH 7.4 and 10.5 for 12 h at 37 °C. Both samples were exchanged against 20 mM Tris HCl, pH 7.4 to neutralize the pH. An amount of 5 µg of each protein sample was analysed independently and as a mixture. The protein samples were mixed with the rehydration solution. An IPG strip of 11 cm, pH range 4–7 pH (GE Healthcare) was rehydrated in this protein containing solution in the dark, for 12 h at 23 °C. Isoelectric focusing (IEF) was performed on Ettan IPGphor 3 using a standard program for 11 cm IPG strip of 4–7 pH range. This was followed by SDS–PAGE. Gels were stained with Coomassie Brilliant Blue R-250 to visualize the protein spots.

**Circular dichroism.** CD was used to probe the structure and stability of WT and mutants of MjGATase.

Far-ultraviolet and near-ultraviolet CD spectra were recorded using Jasco J-810 spectropolarimeter (Jasco Corporation, Japan) equipped with a Peltier heating system. The protein concentration used for far-ultraviolet CD and near-ultraviolet CD was 10 and 50 µM, respectively. Spectra were recorded in a cuvette of path length 0.1 cm for far-ultraviolet and 1 cm for near-ultraviolet CD. Each far-ultraviolet CD spectrum is an average of three accumulations while each near-ultraviolet CD spectrum is an average of 30 accumulations. Each spectrum was recorded at a scan speed of 50 nm min$^{-1}$. All spectra were corrected for background by subtracting the spectrum of the buffer components from the spectrum of the protein sample.

Thermal unfolding of WT and mutants of MjGATase was monitored by CD at a fixed wavelength of 220 nm and with a temperature ramp of 0.5 °C min$^{-1}$ over the temperature range of 70–100 °C. It should be noted that the CD spectra at 25 and 70 °C were identical and therefore, thermal unfolding was monitored over the range of 70–100 °C. Complete far-ultraviolet CD spectra were also recorded at different temperatures. All spectra were recorded at a protein concentration of 10 µM in 5 mM Tris HCl, pH 7.4, in a cuvette of 0.1 cm path length. All spectra were corrected for background by subtracting the spectrum of the buffer components from the spectrum of protein sample.

Effect of chaotrope on the tertiary structure of WT and mutants was monitored by near-ultraviolet CD. Protein solutions of concentration 50 µM were incubated with 5 M guanidinium chloride (GdmCl) at 37 and 70 °C for 12 h before recording CD spectra. All spectra were recorded in a cuvette of 1 cm path length with a scan speed of 50 nm min$^{-1}$ and data interval of 0.1 nm. Each spectrum was an average of 30 accumulations. All spectra were corrected for background by subtracting the spectrum of the buffer components from that of the protein.

**Fluorescence spectroscopy.** The structural stability of WT and mutants of MjGATase was also probed by fluorescence spectroscopy. Intrinsic tryptophan fluorescence spectra were recorded on Hitachi F-2500 fluorimeter (Hitachi High Technologies, Japan). A protein concentration of 10 μM in 20 mM Tris HCl, pH 7.4 was used with excitation at 290 nm. Excitation and emission slit widths were set at 5 and 10 nm, respectively. All spectra were corrected for background by subtracting the spectrum of the buffer components from the spectrum of the protein sample.

GdmCl induced unfolding of WT and mutants of MjGATase was monitored through fluorescence spectroscopy. A concentration of 10 μM of protein in 20 mM Tris HCl, pH 7.4 was preincubated for 12 h at 37 °C in varied concentrations of GdmCl solution prior to the acquisition of emission spectra. Protein samples were excited at 290 nm. Excitation and emission bandwidths were set at 5 and 10 nm, respectively. All spectra were corrected for background by subtracting the spectrum of the buffer components from the spectrum of the protein sample.

**Size-exclusion chromatography.** Analytical size-exclusion chromatography was performed on a Superdex 200 column ($1 \times 30$ cm) attached to an AKTA Basic HPLC system (GE Health Care Life Sciences, UK). The runs were performed at a flow rate of 0.5 ml min$^{-1}$ and protein elution was monitored simultaneously at 280 and 220 nm. The column was calibrated with β-amylase (200 kDa), alcohol dehydrogenase (150 kDa), bovine serum albumin (66 kDa), carbonic anhydrase (29 kDa) and cytochrome C (12.4 kDa) as molecular weight standards (Sigma Aldrich, USA). A 30 μM protein sample was preincubated in 8 M GdmCl solution for 12 h at 37 °C prior to chromatographic runs. A 100 μl protein sample was injected onto the column pre-equilibrated with 20 mM Tris HCl, pH 7.4 containing 6 M GdmCl.

**Enzyme assays.** MjGMPS is a two-subunit enzyme comprised of MjGATase and MjATPPase. MjGMPS converts XMP to GMP through utilizing the ammonia generated from the hydrolysis of glutamine by MjGATase. MjGATase does not hydrolyse glutamine, independent of MjATPPase. Therefore, MjGATase activity was monitored by coupling with MjATPPase using Hitachi U2010 UV-Vis spectrophotometer fitted with water-circulated cell holder (Hitachi High Technologies, Japan). GMP formation from XMP by the utilization of NH$_3$ generated from glutamine hydrolysis by MjGATase was monitored by measuring the continuous decrease in absorbance at 290 nm. Conversion of XMP ($\varepsilon_{290} = 4,800$ M$^{-1}$ cm$^{-1}$) to GMP ($\varepsilon_{290} = 3,300$ M$^{-1}$ cm$^{-1}$) that leads to a drop in $\Delta\varepsilon_{290}$ value of 1,500 M$^{-1}$ cm$^{-1}$ was used to estimate the concentration of GMP formed[61]. 10 μM of enzyme (WT and the mutants of MjGATase) in 20 mM Tris HCl, pH 7.4 was preincubated for 30 min at varying temperatures. The time period of preincubation at 100 °C was 15 min. Samples were centrifuged at 16,000$g$ for 15 min and supernatant was used for activity measurement at 70 °C. The assay mix consisted of 90 mM HEPES, pH 7.0, 3 mM ATP, 200 μM XMP, 20 mM MgCl$_2$ and 5 mM glutamine in a total volume of 250 μl. The reaction was initiated by addition of *in vitro* reconstituted wild-type MjGATase (or mutants) and MjATPPase in equimolar concentration of 1 μM each.

**Data availability.** The data that support the findings of this study are available within the article and its supplementary files and from the corresponding author upon request.

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

## Acknowledgements

H.B. acknowledges Department of Biotechnology, Government of India for funding. S.K. thanks the Jawaharlal Nehru Centre for Advanced Scientific Research, Government of India for a postdoctoral fellowship. S.K. and K.G. acknowledge the Council of Scientific & Industrial Research, Government of India for junior and senior research fellowships. We acknowledge Professor Carol Robinson for the use of the Orbitrap MS and Dr Joseph Gault and Stephen Ambrose for their help with the instrument. K.G. is currently a fellow of the Royal Commission for the Exhibition of 1851, a Junior Research Fellow at the St Catherine's College, Oxford and recipient of the John Fell Research grant. We thank Professor Utpal S. Tatu, Department of Biochemistry, Indian Institute of Science, Bangalore for permitting us the use of Ettan IPGphor 3. We acknowledge Sharanya Chaterjee, a graduate student at the Indian Institute of Science for her help in performing IEF.

## Author contributions

All experimental studies were carried out by S.K. Mass spectra were recorded by S.P. and K.G. A.D. helped S.K. in the construction of mutants. Mass spectrometry data were analysed by S.K., S.P., P.B., H.B. and K.G. The manuscript was written by S.K., H.B. with inputs from P.B. All authors read and approved the final manuscript.

## Additional information

**Competing financial interests:** The authors declare no competing financial interests.

