## [Peer review file · Nature Communications]

Reviewers' Comments:

Reviewer #1 (Remarks to the Author)

Summary of the key results

Kumar et al. shows by a large variety of techniques (mass spectrometry, mutagenesis, thermal denaturation, equilibrium unfolding, size exclusion chromatography and activity measurements) that the existence of a succinimide at position 109 in the glutaminase domain of *M. jannaschii* GMP synthetase is essential for the structural integrity and optimal functioning of the enzyme at the high temperatures encountered physiologically by this hyperthermophilic organism. Succinimide is extremely stable as it resists high temperatures (100°C), presence of chaotrope (8M guanidine) and low pH (pH 2.5).

Moreover, the same techniques were used to show that the following residue (D110) was crucial for stabilizing the succinimide (it is formed in the D110G and D100K mutants at acidic pH but rapidly hydrolyzed at neutral pH) as well as the structure of the protein.

Originality and interest

The role of the presence of a succinimide in imparting structural stability to a protein has no precedent in the literature. The results highlight a new mechanism for thermostability and are counterintuitive to the well-known mechanism of deamidation, which is one of the potential causes for loss of protein structure at elevated temperatures. Other succinimides that are present in flexible loops of other proteins from thermophilic organisms could also impart rigidity to the proteins by a similar mechanism.

A Data & methodology: validity of approach, quality of data, quality of presentation

B Appropriate use of statistics and treatment of uncertainties

C Conclusions: robustness, validity, reliability

The various methods used are appropriate and the data obtained are sound. Nevertheless, I have a few suggestions to improve the presentation of the manuscript in order to be accessible to non specialists of MS techniques (see major modification).

D Suggested improvements: experiments, data for possible revision

No experiments needed

E References: appropriate credit to previous work? Yes

Clarity and context: lucidity of abstract/summary, appropriateness of abstract, introduction and conclusions

Major modification

One of the main techniques used in this manuscript is mass spectrometry and Nature Communications is a journal of broad scope. Therefore, a figure should explain the different fragmentation patterns of a peptide by MALDI, CID and ETD (p.5 lines 12-13). In particular, as differentiation by ETD-MS/MS of Asp and isoAsp is quite subtle, it is important to include a scheme of the fragmentation to understand how the c+57 and z-57 ions are evidence of the isoAsp amino acid (p.5, 3 lines before the end).

Figures 1c and 1d, Figures 2c and 2d: The labels for the z-ions are incorrectly placed under the corresponding fragments.

Supplementary Figure 1b, 1c, 1d, 1e, Supp Figures c, d, e, f: The labels for the y-ions are incorrectly placed under the corresponding fragments

Minor modifications

Revise punctuation throughout the text

p. 4 line 14 "would reduce" instead of "to reduce"
p.4 line 20 "indispensable"
p. 5 line 14 "ETD-MS/MS of the 1,352.7 Da peptide showed c7-17 and z5-17 masses..." and highlight these fragments in Fig. 1c.
p.5 line 15: which ions in the MALDI and CID-MS/MS of the 1370.7 Da peptide indicate the presence of the succinimide? (Sup. Fig 1d, e)
p.5 2 last lines: replace c+57 and z-57 by c6+57 and z5-57 and highlight these fragments in Fig. 1d
p.7 6 lines before the end: c6+57 and z5-57 fragment ions
p.8 6 lines before the end "indicate" instead of "indicates"
p.8 3 lines before the end, to avoid redundancy, change to "was followed by tryptophan fluorescence." and delete "revealed that the enzyme...above 4-5M GdmCl."
p.9 after "in 8M GdmCl." add: "These three mutants possessing no or little succinimide start unfolding above 4-5M GdmCl. In contrast, largely similar..."
p.11 line 5: It is not clear that both protein species contain hydrolyzed succinimide. It is better to write : "presence of hydrolyzed succinimide in two species: wild type protein as well as oxidized protein containing two additional oxygen atoms"
p.17 Legend Figure 3
1st line (a) change to: "Molar ellipticity recorded at 220nm using a CD spectrophotometer as a function..."

p.18 Legend Figure 5
1st line: (a) change title to: Fluorescence emission maximum versus pH.
3rd line: [B] (Mobs 21,035) (not 21,033).

Fig. 2c: peptide within the figure is lacking L-F-K
Fig. 4 should be moved to Supplementary Figures (combine with Supp Figure 6).

Reviewer #2 (Remarks to the Author)

A. Summary of the key results

This work has two key aspects:

- A.1. The "unusual" stability of aspartyl succinimide, e.g., resistance to hydrolysis at high temperature (100 C);
- A.2. Stabilizing effects of the succinimide on the protein structure, e.g., resistance to denaturation at 6 M guanidinium.

B. Originality and interest: if not novel, please give references

B.1. "Unusual" stability of succinimide: as the authors commented, succinimide has been considered labile by many in the field. On the other hand, its stability has also been well documented and recognized. For instance, many bioconjugates contain succinimide thioether resulting from maleimide-thiol reaction. And also noted by the authors, succinimide has been observed in other proteins (see additional examples in Ouellette 2013 and Klaene 2013), albeit not full conversion. Finally, the fact that succinimide in the tryptic peptides was routinely observed in this work clearly demonstrates the relative stability of succinimide. So it is worth addressing some additional questions, such as (1) how much more stable and (2) why.

B.2. The stabilizing effects of the succinimide on the protein structure are noticeable, e.g., by comparing to the mutants that cannot form succinimide. Of course, any mutation is likely to change the structure, perhaps significantly, especially for a well evolved protein. This reviewer is not familiar with the folding of thermophilic proteins, so it would be helpful if the authors could put this into perspective. For example, are the changes in stability observed in this work (WT vs mutants) significantly more than other thermophilic proteins?

B.3. The stability of succinimide and protein is coupled to each other. As such, a question this

reviewer has is whether the stability of succinimide is due to shielding of water (nucleophile for hydrolysis) by the protein. This perhaps can be tested by treating the protein with stronger nucleophiles such as hydroxylamine or hydrazine (see Zhu 2007 and Klaene 2013) under near neutral pH.

B.4. Mechanism and kinetics of succinimide formation. These aspects are not discussed in great details but perhaps are more interesting. For instance, it is noted that re-formation of succinimide from the hydrolysis products were slow in vitro. This raises an interesting possibility that in vivo formation of succinimide may be catalyzed, say by additional enzymes or other factors.

On a related note, the full experimental details for the expression and purification should be included instead of referring to previous papers. This will allow the readers to see how the conditions may contribute to succinimide formation. For example, whether the non-succinimide species were removed during purification.

C. Data & methodology: validity of approach, quality of data, quality of presentation

C.1. Overall, the experiments were well designed and data were of high quality.

D. Appropriate use of statistics and treatment of uncertainties

D.1. Not applicable.

E. Conclusions: robustness, validity, reliability

E.1. The findings (data) are robust. As mentioned above, the key is how significant the findings are (e.g., how much more stable and why).

F. Suggested improvements: experiments, data for possible revision

F.1. Also provide ESI mass spectra after deconvolution. Easier to see the mass changes.

F.2 To provide mechanistic insight and fully assess the unusual factors, it would be helpful to chemically synthesize authentic tryptic peptides and examine their kinetics to form succinimide and the stability of the resulting succinimide.

F.3. The authors touched upon minimizing artifacts of asparaginyl deamidation and aspartyl dehydration during sample preparation, which should be explicitly discussed and addressed. Some recent methods to monitor (e.g., ¹⁸O labeling, see Du 2012 and Liu 2012) and eliminate (e.g., Glu-C digestion at pH 4, see Liu 2016) such artifacts should be discussed, and if needed, implemented. Again to better assess the potential artifacts, full experimental details for tryptic digestion should be included.

F.3. Formation of succinimide can lead to changes in pI, which may be readily detected by isoelectric focusing (IEF). An additional advantage is that IEF is mostly independent of protein conformation. Such data may tease apart the effects of protein structure from the effects of chemical transformation.

F.4. In supplementary Figure 1 and other places, MALDI-MS/MS and CID-MS/MS are mentioned. Not clear what MALDI-MS/MS is exactly. ETD-MS/MS? Please clarify.

F.5. In supplementary Figure 5, on my screen, the blue color is too dark to tell from black.

G. References: appropriate credit to previous work?

Below are some additional relevant references should be cited.

Selective cleavage of isoaspartyl peptide bonds by hydroxylamine after methyltransferase priming.

Zhu JX, Aswad DW.

Anal Biochem. 2007 May 1;364(1):1-7. Epub 2007 Feb 22.

PMID: 17376395

Determination of deamidation artifacts introduced by sample preparation using ¹⁸O-labeling and tandem mass spectrometry analysis.

Du Y, Wang F, May K, Xu W, Liu H.

Anal Chem. 2012 Aug 7;84(15):6355-60. doi: 10.1021/ac3013362. Epub 2012 Jul 17.

PMID: 22881398

Protein isoaspartate methyltransferase-mediated 18O-labeling of isoaspartic acid for mass spectrometry analysis.

Liu M, Cheetham J, Cauchon N, Ostovic J, Ni W, Ren D, Zhou ZS.

Anal Chem. 2012 Jan 17;84(2):1056-62. doi: 10.1021/ac202652z. Epub 2011 Dec 27.

PMID: 22132761

Comparison of the in vitro and in vivo stability of a succinimide intermediate observed on a therapeutic IgG1 molecule.

Ouellette D, Chumsae C, Clabbers A, Radziejewski C, Correia I.

MAbs. 2013 May-Jun;5(3):432-44. doi: 10.4161/mabs.24458. Epub 2013 Apr 22.

PMID: 23608772

Detection and quantitation of succinimide in intact protein via hydrazinetraping and chemical derivatization.

Klaene JJ, Ni W, Alfaro JF, Zhou ZS.

J Pharm Sci. 2014 Oct;103(10):3033-42. doi: 10.1002/jps.24074. Epub 2014 Jul 14.

PMID: 25043726

Mildly acidic conditions eliminate deamidation artifact during proteolysis: digestion with endoprotease Glu-C at pH 4.5.

Liu S, Moulton KR, Auclair JR, Zhou ZS.

Amino Acids. 2016 Jan 9. [Epub ahead of print]

PMID: 26748652

H. Clarity and context: lucidity of abstract/summary, appropriateness of abstract, introduction and conclusions

H.1. Graphic abstract is bit too crowded and fonts are a bit too small. Suggest to simplify.

Reviewer #3 (Remarks to the Author)

This paper describes an intriguing finding of a remarkably stable succinimide in an enzyme from a hyperthermophilic archaeon. This is contrary to the belief that succinimide is transiently formed as an intermediate during asparaginyl deamidation or aspartyl dehydration in proteins, and its formation is followed by rapid hydrolysis of this intermediate to aspartyl and isoaspartyl residue. A significant part of the evidence is based on mass spectrometric data presented in Figures 1, 2, 4, and 5, as well as Supplementary Figures 1, 2, 3, 4, 6 and 7. As this reviewer is an expert in mass spectrometry, these data drew most of his attention.

The mass spectrometry data please with their abundance, but are not without flaws. These potential flaws are discussed below (not necessarily in the order of significance).

1. First off, the protein spectra (Fig. 1b, e, f; 2 a, b, e, f; etc.) are all taken with different signal-to-noise ratios, ranging from ≈ 5 -6 in Fig. 1e to ≈ 20 in Fig. 2b. This could mean that the spectra were taken at different source conditions (temperature, nozzle-skimmer voltage, source cleanliness, etc.). Often, spectra taken at lower S/N show a single peak (e.g., Fig. 1e, 2a), or a dominant peak at a large m/z (Fig. 5c), while those taken at higher S/N show either plurality of peaks (e.g., 1f, 2b and f), or dominant peaks at lower m/z (e.g., 2e, 5b). Could that be an artifact of different source conditions? It is well known in mass spectrometry, that depending upon source conditions, one can obtain protein spectra with different extent of small-molecule losses, such as NH₃ and H₂O loss, i.e., -17 and -18 Da, respectively. These artificial losses would be indistinguishable in mass from the mass defect due to the succinimide presence.

2. Adding to the above suspicion, the insets are shown with different magnification of the m/z scale, and the latter is never given. The difference in scales (e.g., compare insets in Fig. 2a and 2b) is probably one order of magnitude. Why would so different scales be needed? Even Supplementary Figure 3 shows different and unspecified scales.

3. Contrary to what is customary in protein mass spectrometry, only one charge state is zoomed in, instead of demonstrating the results of neutral mass deconvolution. That is not surprising if the peak ratios change dramatically with the charge state, as would be in the case of gas-phase losses (artifacts), which would be stronger from higher charge states.

4. In general, the mass spectrometric resolution is quite low. Many of today's instruments, including qTOFs, Orbitraps and FT ICR MS can easily resolve isotopic peaks of a 23 kDa protein. With such isotopic resolution, the difference between the NH_3 and H_2O losses would be much more clear. Why hasn't been high resolution used in at least most important cases?

5. It is great that the authors used a plurality of MS/MS techniques, including CID and ETD, but the results are not always consistent with their conclusions. It is amazing that the peptide in Supplementary Fig.1a has survived harsh MALDI conditions while preserving its succinimide. Equally remarkable is that this peptide has apparently survived also electrospray ionization to produce an ETD MS/MS spectrum in Fig. 1c. The authors don't comment on this apparent stability of succinimide in a peptide. If it is so stable, why not perform LC-MS/MS analysis and quantify peptide abundances as common in proteomics instead of relying on low-resolution protein mass spectra?

6. In all MS/MS spectra, the C-terminal series of fragments (γ - and z -) appear to be mislabeled, with a series starting from z_2 or γ_2 ions that have only one amino acid, K.

These issues need to be addressed before the paper could be accepted for publication.

Point-to-point response to the reviewers

Reviewer 1

Summary of the key results

Kumar et al. shows by a large variety of techniques (mass spectrometry, mutagenesis, thermal denaturation, equilibrium unfolding, size exclusion chromatography and activity measurements) that the existence of a succinimide at position 109 in the glutaminase domain of M. jannaschii GMP synthetase is essential for the structural integrity and optimal functioning of the enzyme at the high temperatures encountered physiologically by this hyperthermophilic organism.

Succinimide is extremely stable as it resists high temperatures (100{degree sign}C), presence of chaotrope (8M guanidine) and low pH (pH 2.5). Moreover, the same techniques were used to show that the following residue (D110) was crucial for stabilizing the succinimide (it is formed in the D110G and D100K mutants at acidic pH but rapidly hydrolyzed at neutral pH) as well as the structure of the protein.

Originality and interest

The role of the presence of a succinimide in imparting structural stability to a protein has no precedent in the literature. The results highlight a new mechanism for thermostability and are counterintuitive to the well-known mechanism of deamidation, which is one of the potential causes for loss of protein structure at elevated temperatures. Other succinimides that are present in flexible loops of other proteins from thermophilic organisms could also impart rigidity to the proteins by a similar mechanism.

The various methods used are appropriate and the data obtained are sound. Nevertheless, I have a few suggestions to improve the presentation of the manuscript in order to be accessible to non specialists of MS techniques (see major modification).

Response to Reviewer 1

#Query1

One of the main techniques used in this manuscript is mass spectrometry and Nature Communications is a journal of broad scope. Therefore, a figure should explain the different fragmentation patterns of a peptide by MALDI, CID and ETD (p.5 lines 12-13). In particular, as differentiation by ETD-MS/MS of Asp and isoAsp is quite subtle, it is important to include a

scheme of the fragmentation to understand how the c+57 and z-57 ions are evidence of the isoAsp amino acid (p.5, 3 lines before the end).

Response

As suggested by this reviewer we have now included a figure (**Figure 2**) in the main text of the revised manuscript that shows pattern of fragmentation by tandem (MS/MS) mass spectrometry. The fragment ion nomenclature and selected techniques for MS/MS fragmentation are also included. Further, a scheme showing fragmentation by ETD leading to the generation of c+57 and z-57 ions in an isoAsp containing peptide has also been added.

#Query2

Figures 1c and 1d, Figures 2c and 2d: The labels for the z-ions are incorrectly placed under the corresponding fragments.

Response

The incorrectly placed labels for z ions in ETD-MS/MS figures have been corrected. We apologize for the error.

#Query3

Supplementary Figure 1b, 1c, 1d, 1e, Supp Figures c, d, e, f: The labels for the y-ions are incorrectly placed under the corresponding fragments

Response

The incorrectly placed labels for y ions in supplementary figures have been corrected.

#Query 4

Minor modifications

Revise punctuation throughout the text

Response

Revised draft has punctuations added/corrected.

p. 4 line 14 "would reduce" instead of "to reduce"-

Response

Corrected (**page 4, line 104**)

p.4 line 20 "indispensable"

Response

Spelling corrected (**page 4, line 110**)

p. 5 line 14 "ETD-MS/MS of the 1,352.7 Da peptide showed c7-17 and z5-17 masses..." and highlight these fragments in Fig. 1c.

Response

These have been highlighted and the legend appropriately modified (**page 6, line 140**)

p.5 line 15: which ions in the MALDI and CID-MS/MS of the 1370.7 Da peptide indicate the presence of the succinimide? (Sup. Fig 1d, e)

Response

MALDI and CID-MS/MS of the 1370.7 Da peptide do not indicate the presence of succinimide. The MS/MS of this peptide by MALDI and CID indicate hydrolysis of succinimide, as evident from the increase of 1 Da in the masses of b₇ to b₁₀ and y₅ to y₁₀ fragment ions (**page 6, lines 143-145**). **Figure 1d** which is the ETD spectrum of this peptide shows the presence of isoAsp at position 109 as evident by the diagnostic C₆+57 and z₅-57 ions, confirming the hydrolysis of succinimide. The presence of succinimide is evident in the data presented in **Figure 1b, c, Supplementary Figure 2b and c**.

p.5 2 last lines: replace c+57 and z-57 by c6+57 and z5-57 and highlight these fragments in Fig. 1d.

Response

This has been corrected in the text (**page 6, line 153**) and these ions are highlighted in the figure (**Figure 1d**)

p.7 6 lines before the end: c6+57 and z5-57 fragment ions-

Response

Corrected (**page 8, lines 206-207**)

p.8 6 lines before the end "indicate" instead of "indicates"

Response

This section has been edited in the revised draft (**pages 9-10, lines 231-239 and pages 11-12, lines 284-290**)

p.8 3 lines before the end, to avoid redundancy, change to "was followed by tryptophan fluorescence." and delete "revealed that the enzyme...above 4-5M GdmCl."

Response

Corrected as instructed (**page 10, line 241**)

p9 after "in 8M GdmCl." add: "These three mutants possessing no or little succinimide start unfolding above 4-5M GdmCl. In contrast, largely similar..."

Response

Corrected (**page 10, lines 249-250**)

p.11 line 5: It is not clear that both protein species contain hydrolyzed succinimide. It is better to write : "presence of hydrolyzed succinimide in two species: wild type protein as well as oxidized protein containing two additional oxygen atoms"

Response

Corrected (**page 13, lines 321-322**)

p.17 Legend Figure

1st line (a) change to: "Molar ellipticity recorded at 220nm using a CD spectrophotometer as a function..."

Response

Corrected (**Legend to figure 4, page 37, lines 975-977**)

p.18 Legend Figure 5

1st line: (a) change title to: Fluorescence emission maximum versus pH.

3rd line: [B] (Mobs 21,035) (not 21,033).

Response

(a) Corrected (**Legend to figure 6, page 38, line 1001**)

(b) The new panel highlights only one species of M_{obs} 21,003 Da, indicating presence of the succinimide intermediate (**Figure 6b**).

Fig. 2c: peptide within the figure is lacking L-F-K

Response

The sequence LFK is not missing. As we have introduced a lysine residue in place of aspartyl at position 110, trypsin cleaves after this lysine (K110). Hence, the peptide fragment obtained from trypsin digestion of MjGATase_D110K lacks LFK residues.

Fig. 4 should be moved to Supplementary Figures (combine with Supp Figure 6).

Response

This is the key data showing the remarkable stability of succinimide. Therefore, we would like to retain it as a figure in the main text of the manuscript. Further, as reviewer 2 has asked for additional data in support of succinimide stability, we have carried out new experiments. These pertain to the examination of succinimide stability in 0.1 N HCl and in the presence of a stronger nucleophile, NH_2OH . These mass spectral data have been added as additional panels to this **Supplementary Figure 8**.

Reviewer 2

Summary of the key results

This work has two key aspects:

A.1. The "unusual" stability of aspartyl succinimide, e.g., resistance to hydrolysis at high temperature (100 C);

A.2. Stabilizing effects of the succinimide on the protein structure, e.g., resistance to denaturation at 6 M guanidinium.

Originality and interest

"Unusual" stability of succinimide: as the authors commented, succinimide has been considered labile by many in the field. On the other hand, its stability has also been well documented and

recognized. For instance, many bioconjugates contain succinimide thioether resulting from maleimide-thiol reaction. And also noted by the authors, succinimide has been observed in other proteins (see additional examples in Ouellette 2013 and Klaene 2013), albeit not full conversion. Finally, the fact that succinimide in the tryptic peptides was routinely observed in this work clearly demonstrates the relative stability of succinimide. So it is worth addressing some additional questions, such as (1) how much more stable and (2) why. B. Originality and interest: if not novel, please give references

B.1. "Unusual" stability of succinimide: as the authors commented, succinimide has been considered labile by many in the field. On the other hand, its stability has also been well documented and recognized. For instance, many bioconjugates contain succinimide thioether resulting from maleimide-thiol reaction. And also noted by the authors, succinimide has been observed in other proteins (see additional examples in Ouellette 2013 and Klaene 2013), albeit not full conversion. Finally, the fact that succinimide in the tryptic peptides was routinely observed in this work clearly demonstrates the relative stability of succinimide. So it is worth addressing some additional questions, such as (1) how much more stable and (2) why.

Response to reviewer 2

This referee raises two key questions: (1) as compared to earlier reports, succinimide in MjGATase is how much more stable? (2) Why it is so stable?

These questions are addressed in detail below.

A.1. The "unusual" stability of aspartyl succinimide, e.g., resistance to hydrolysis at high temperature (100 C);

Response

The succinimide in MjGATase is formed by loss of NH₃ from an asparaginyl residue (N109) and not an aspartyl residue. However, we have also observed aspartyl succinimide in MjGATase_N109D

B.1. "Unusual" stability of succinimide: as the authors commented, succinimide has been considered labile by many in the field. On the other hand, its stability has also been well documented and recognized. For instance, many bioconjugates contain succinimide thioether

resulting from maleimide-thiol reaction. And also noted by the authors, succinimide has been observed in other proteins (see additional examples in Ouellette 2013 and Klaene 2013), albeit not full conversion. Finally, the fact that succinimide in the tryptic peptides was routinely observed in this work clearly demonstrates the relative stability of succinimide. So it is worth addressing some additional questions, such as (1) how much more stable and (2) why.

Response

The referee correctly raises the issue of the significantly greater tendency of succinimide in proteins to be hydrolyzed as compared to the aqueous stability of the imide intermediate, noted by researchers in the field of bioconjugates. A corollary to this referee's point would be a question as to why succinimidyl residues in proteins are intrinsically so unstable as compared to the thioether analog form in maleimidethiol conjugate. It is likely that the inherent susceptibility of succinimides to hydrolysis, when incorporated in protein structures (as compared to the relative stability of succinimide moiety in bioconjugate chemistry as pointed out by the reviewer), is due to the steric strain imposed in the context of the protein backbone (**page 15, lines 370-374**).

In response to the referee's first point regarding the stability of succinimide in MjGATase, we would like to highlight that complete conversion of aspartyl/asparaginyl residue to succinimidyl intermediate has rarely been observed in proteins and stability of this intermediate at extremely high temperatures and in the presence of high concentrations of chaotrope has no precedence. This is the first study that demonstrates the existence of a remarkably stable succinimide. The imide intermediate in MjGATase remains fully intact even at 100 °C, in the presence of 8 M GdmCl, 0.1 N HCl and in the presence of high concentrations of a stronger nucleophile (2 M NH₂OH). These findings strongly support the unprecedented stability of succinimide in this hyperthermophilic enzyme. Succinimide stability measurements in 0.1 N HCl and 2 M NH₂OH have been added to the revised draft (**Supplementary Figure 8c and d and page 12, lines 300-308**).

In response to the referee's second point as to why succinimide is so stable in MjGATase, we note that the structure of MjGATase appears to play a key role in protecting succinimide from hydrolysis. Rapid hydrolysis of succinimide in the unfolded protein at high pH (**Figure 6c**), in tryptic peptides (**Supplementary Figure 2a**) and D110 mutants (**Figure 3**) strongly supports

the role of protein structure in shielding the succinimide from hydrolysis. Further, as suggested by this reviewer we have also examined the potential of the synthetic peptide VYVDKENDLFK to form and retain succinimide. The peptide was found not to form the succinimide.

(**Supplementary Figure 11**). This aspect has been brought out on **page 14, lines 343-357**.

In response to the referee's point on the detection of succinimide moiety in the tryptic peptide, it should be noted that it is present only in relatively minor amounts with hydrolysis being more facile in the cleaved product (**Supplementary Figure 2a**). The **line 156 on page 6** has been corrected to emphasize that the succinimide containing peptide in the tryptic digest is present only in small amounts. Further, it should be noted that the trypsin digestion was performed at pH 7.0 to preserve the succinimide moiety in the tryptic fragment (**methods section; pages 22-23, lines 556-578**).

B.2. The stabilizing effects of the succinimide on the protein structure are noticeable, e.g., by comparing to the mutants that cannot form succinimide. Of course, any mutation is likely to change the structure, perhaps significantly, especially for a well evolved protein. This reviewer is not familiar with the folding of thermophilic proteins, so it would be helpful if the authors could put this into perspective. For example, are the changes in stability observed in this work (WT vs mutants) significantly more than other thermophilic proteins?

Response

Significantly, the absence of succinimidyl moiety in the mutants (MjGATase_N109S, D110G and D110K) results in a T_m of 85-90 °C (**Fig. 4a**). This relatively high T_m suggests that as, in other thermophilic proteins the sequence of *Methanocaldococcus jannaschii glutaminase* (MjGATase) confers a high degree of thermostability. The spontaneous formation of succinimide in the wild-type enzyme results in a dramatic enhancement of melting temperature, conferring hyperthermostability as evident by the absence of melting even at 100 °C. This aspect is described **on pages 9, lines 234-239**.

In response to the referee's point on the effect of mutation on the protein stability we would like to highlight that, mutations that perturb the interactions (hydrophobic interactions and salt bridges) in the core of the protein would be expected to alter the structure significantly¹. However, in MjGATase the asparaginy residue (N109) that forms the succinimide is present in a loop that is on the surface of the protein structure. Our studies suggest that the succinimide

stabilizes the protein structure possibly; by imparting rigidity through side-chain-backbone cyclization that confines the loop movements (**page 18, lines 437-439**). This is evident from studies of the mutants (MjGATase_N109S, D110G and D110K) that lack succinimide and exhibit reduced stability (**Fig. 4a and b**). This inference is further corroborated by the stability of the MjGATase_N109D mutant where a change in the residue side chain from an amide (Asn) to an acid (Asp) has not destabilized the protein as both residues convert to a stable succinimide (**Fig. 4a and b**).

We observed that mutations (MjGATase_N109S, D110K and D110G) that do not enable side-chain-backbone cyclization destabilize the enzyme whereas a non-conserved mutation (N109D) that retains this ability is as stable as the WT protein (**Fig. 4**). Therefore, it is safe to conclude that the difference in the stabilities of mutants, lacking a stable succinimide and those (WT_{Su} MjGATase and N109D_{Su}) harboring the intermediate is a result of side-chain-backbone cyclization and not due to mutations per se (**pages 11-12, lines 284-290**).

Thermophilic enzymes exploit a variety of tools to enhance structure stability, with none being universal¹. This is the first example wherein a stable succinimide enables the enzyme to retain its structure even at 100 °C or in 8 M GdmCl, a feature not reported earlier.

B.3.*The stability of succinimide and protein is coupled to each other. As such, a question this reviewer has is whether the stability of succinimide is due to shielding of water (nucleophile for hydrolysis) by the protein. This perhaps can be tested by treating the protein with stronger nucleophiles such as hydroxylamine or hydrazine (see Zhu 2007 and Klaene 2013) under near neutral pH.*

Response

Yes indeed, our findings strongly suggest that the protein structure protects the succinimide from hydrolysis by shielding the imide intermediate from bulk water. This aspect has been discussed on **page 16, lines 393-396, 405-410**. The role of protein structure in shielding succinimide from hydrolysis is clearly evident by the facile hydrolysis of this intermediate in the tryptic peptide (**Supplementary Figure 2a**), in unfolded protein (**Figure 6c**) and D110 mutants (**Fig. 3**). As suggested by this reviewer, we have examined the stability of the succinimide to NH₂OH (**Supplementary Figure 8d**). An appreciable amount of the succinimide modification was present even after incubation for 2 hr at 37° C in a solution containing 2 M NH₂OH.

(**Supplementary Figure 8d and page 12, lines 301-308**). The appearance of additional species after extensive exposure to NH_2OH may be assigned to either hydroxyimic acid derivative or oxidation of methionine residues present in the protein. However, peptide chain cleavage as reported in earlier studies on peptides and proteins²⁻⁴ was not observed in MjGATase, even after long exposure to NH_2OH . The role of protein structure on the stability of the succinimidyl residue is well brought out by the absence of this modification in the synthetic peptide.

B.4. Mechanism and kinetics of succinimide formation. These aspects are not discussed in great details but perhaps are more interesting. For instance, it is noted that re-formation of succinimide from the hydrolysis products were slow in vitro. This raises an interesting possibility that in vivo formation of succinimide may be catalyzed, say by additional enzymes or other factors.

On a related note, the full experimental details for the expression and purification should be included instead of referring to previous papers. This will allow the readers to see how the conditions may contribute to succinimide formation. For example, whether the non-succinimide species were removed during purification.

Response

The referee notes slower rate of succinimide re-formation from the hydrolyzed product as an interesting phenomenon and suggests the role of additional enzymes or other factors for succinimide formation *in vivo*. In the cell, succinimide formation from isoAsp catalyzed by protein L-isoaspartyl methyl transferase (PIMT) mediates the repair of isoAsp to Asp through succinimidyl intermediate⁵ (**page 17, lines 425-426**).

It should be noted that hydrolysis of succinimide in MjGATase is achieved only after prolonged exposure (12 hrs at 37 °C) at high pH (10.5) (**Figure 6c**). We note that under these conditions the enzyme is also unfolded (**Figure 6a and d**). The slower rate of succinimide formation from the hydrolyzed product (Asp/isoAsp) in refolded MjGATase could be attributed to the absence of complete reversion to the native fold or presence of high amounts of isoAsp in hydrolyzed product or a combination of both (**pages 13-14, lines 334-342**). The near complete conversion of the hydrolyzed product to succinimide in MjGATase_D110G which was not unfolded by pre-exposure to high pH supports the above inference (**Fig.3e**). *In vitro* slower rate

of succinimide formation from isoASP as compared to aspartyl or asparaginyl residue has been reported earlier⁶.

As suggested by the reviewer the full experimental details for the expression and purification of MjGATase has been included in the revised manuscript (**methods section; pages 19-20, lines 469-494**) Briefly, the purification procedure involves incubation of the cell lysate at 70 °C to remove the thermolabile *E. coli* proteins. It is unlikely that this step would remove non-succinimide proteins as the mutants containing little or no succinimide are fully stable and remain soluble at 70 °C. After thermal precipitation, the supernatant was subjected to anion exchange chromatography. A salt (NaCl) gradient was used to elute the protein from the column. Wild type and mutants of MjGATase eluted as a single peak from the column and all fractions under this peak were pooled. Mass spectrum of the protein from this peak, for the wild type and N109S mutant always yielded a single unique mass. The mass spectrum of MjGATase_D110G, which also elutes as a single peak on anion exchange chromatography shows the presence of 2 species; one with succinimide and the other, the hydrolyzed product. Similarly, MjGATase_N109D elutes as a single peak on anion exchange chromatography but shows the presence of two species; a major species with succinimide and a small fraction of native protein. This clearly indicates that under our purification conditions, the hydrolyzed product or the protein containing the precursor sequence, if present, are not separated. This also supports our conclusion that the wild type protein is only in the form of an extremely stable succinimide. Further, as purification of MjGATase involved a step of heating and anion-exchange chromatography, an N-terminal (His)₆-tagged MjGATase was generated and the recombinant protein was directly purified from the cell lysate using Ni-NTA affinity chromatography(**methods section; pages 20-21, lines 495-519**) pages to rule out artifacts of purification procedure leading to formation or removal of succinimide. Fractions containing MjGATase, when pooled and examined by ESI-MS also showed loss of 17 Da (**Supplementary Fig.1**), suggesting that the purification procedure does not contribute to the formation of succinimide or removal of non-succinimide species. This has been added to the revised draft, **page 5, lines 119-126**).

#F.1. Also provide ESI mass spectra after deconvolution. Easier to see the mass changes.

Response

Deconvoluted ESI mass spectra are provided in the revised manuscript.

F.2 To provide mechanistic insight and fully assess the unusual factors, it would be helpful to chemically synthesize authentic tryptic peptides and examine their kinetics to form succinimide and the stability of the resulting succinimide.

Response

As suggested by this reviewer an eleven residue peptide (VYVDKENDLKF), corresponding to the tryptic fragment (V₁₀₃YVDKENDLKF₁₁₃) was synthesized (**Supplementary Figure 11a**) and examined for its ability to form succinimide. However, neither the succinimide nor the hydrolyzed product was detected in the synthetic peptide (**Supplementary Figure 11b-d**). The complete absence of succinimide intermediate in the synthetic peptide under the conditions where full-length enzyme shows complete conversion of N109 to succinimide strongly supports the role of protein structure in enabling spontaneous succinimide formation and stabilization of the intermediate. This has been included in the revised manuscript (**Supplementary Figure 11 and page 14, lines 343-357**).

F.3a. The authors touched upon minimizing artifacts of asparaginyl deamidation and aspartyl dehydration during sample preparation, which should be explicitly discussed and addressed. Some recent methods to monitor (e.g., 18O labeling, see Du 2012 and Liu 2012) and eliminate (e.g., Glu-C digestion at pH 4, see Liu 2016) such artifacts should be discussed, and if needed, implemented. Again to better assess the potential artifacts, full experimental details for tryptic digestion should be included.

Response

Under all conditions of purification and MS recording, wild-type enzyme shows loss of 17 Da while N109S mutant exhibits expected mass. This shows that the deamidation observed is not an artifact arising from sample preparation. As suggested by the reviewer full experimental details of trypsin digestion and peptide extraction have been elaborately written in the methods section of the revised manuscript (**methods section; pages 22-23, lines 556-578**).

#F.3b. Formation of succinimide can lead to changes in pI, which may be readily detected by isoelectric focusing (IEF). An additional advantage is that IEF is mostly independent of protein conformation. Such data may tease apart the effects of protein structure from the effects of chemical transformation.

Response

We agree with the reviewer that formation of succinimide can lead to changes in pI, which may be readily detected by isoelectric focusing (IEF). However, this would be indeed true only if, succinimide was formed from an aspartyl residue in a protein where a unit charge difference may be anticipated between the two forms of protein. In the present case, the succinimide formation takes place at a neutral asparaginyl residue, resulting in no net charge difference between the two forms. The clearest distinction between asparaginyl and succinimidyl form would be evident in their masses. Indeed, our mass spectral studies on intact proteins, tryptic fragments in conjunction with site-directed mutagenesis strongly suggest the transformation of N109 to succinimidyl form. However, as suggested by the reviewer, we performed the isoelectric focusing. IEF followed by SDS-PAGE of MjGATase after preincubation at different pH (7.4 and 10.5) also showed that high pH yields a protein species with a pI distinct from that of the native sample (incubated at pH 7.4) (**Supplementary Fig.10**). The lower pI value of the protein sample incubated at pH 10.5 indicates hydrolysis of succinimide to Asp/isoAsp. The IEF results corroborate mass spectrometry and native-PAGE results (**Fig.3c, Fig.6c and d**). This result has been added to the revised draft (**page 13, lines 329-333**).

#F.4. In supplementary Figure 1 and other places, MALDI-MS/MS and CID-MS/MS are mentioned. Not clear what MALDI-MS/MS is exactly. ETD-MS/MS? Please clarify.

Response

The abbreviations, MALDI, CID and ETD have been expanded on **page 6 lines 137-138**. MS/MS corresponds to tandem mass spectrometry that enables fragmentation of the parent peptide by different methods such as CID, ETD and through MALDI (MS/MS) (**Figure 2**). The legend to Figure 2 has “tandem (MS/MS) mass spectrometry” included. Supplementary Figure 1b (**now reads 2b**) is MALDI-MS/MS and Figure 1c (**now reads 2c**) is the CID MS/MS of 1352.7 Da ($V_{103}YVDKEN_{109}SuDLFK_{113}$) peptide fragment derived from the

trypsin digestion of WT MjGATase. This peptide shows loss of NH₃ from N109. Similarly, Supplementary Figure 1d (**now reads 2d**) is MALDI MS/MS and 1e (**now reads 2e**) is CID-MS/MS of 1370.7 Da (V₁₀₃YVDKED/isoD₁₀₉LFK₁₁₃) tryptic fragment of WT MjGATase that show hydrolysis of succinimide at position 109. Appropriate legends for ETD-MS/MS have been provided in the revised manuscript.

F.5. In supplementary Figure 5, on my screen, the blue color is too dark to tell from black.

Response

Blue color is changed to cyan color which is distinct from black in the revised draft (**Supplementary Figure 5 now reads as Supplementary Figure 7**).

G. References: appropriate credit to previous work?

Below are some additional relevant references should be cited.

Selective cleavage of isoaspartyl peptide bonds by hydroxylamine after methyltransferase priming. Zhu JX, Aswad DW. Anal Biochem. 2007 May 1;364(1):1-7. Epub 2007 Feb 22. PMID: 17376395

Determination of deamidation artifacts introduced by sample preparation using 18O-labeling and tandem mass spectrometry analysis. Du Y, Wang F, May K, Xu W, Liu H. Anal Chem. 2012 Aug 7;84(15):6355-60. doi: 10.1021/ac3013362. Epub 2012 Jul 17. PMID: 22881398

Protein isoaspartatemethyltransferase-mediated 18O-labeling of isoaspartic acid for mass spectrometry analysis.

Liu M, Cheetham J, Cauchon N, Ostovic J, Ni W, Ren D, Zhou ZS.

Anal Chem. 2012 Jan 17;84(2):1056-62. doi: 10.1021/ac202652z. Epub 2011 Dec 27. PMID: 22132761

Comparison of the in vitro and in vivo stability of a succinimide intermediate observed on a therapeutic IgG1 molecule.

Ouellette D, Chumsae C, Clabbers A, Radziejewski C, Correia I.

MAbs. 2013 May-Jun;5(3):432-44. doi: 10.4161/mabs.24458. Epub 2013 Apr 22. PMID: 23608772

Detection and quantitation of succinimide in intact protein via hydrazinetrappping and chemical derivatization.

Klaene JJ, Ni W, Alfaro JF, Zhou ZS.

J Pharm Sci. 2014 Oct;103(10):3033-42. doi: 10.1002/jps.24074. Epub 2014 Jul 14.

PMID: 25043726

Mildly acidic conditions eliminate deamidation artifact during proteolysis: digestion with endoproteaseGlu-C at pH 4.5.

Liu S, Moulton KR, Auclair JR, Zhou ZS.

Amino Acids. 2016 Jan 9. [Epub ahead of print] PMID: 26748652

Response

We have included the references that are relevant and relate to this study.

1. Blodgett, J. K., Loudon, G. M. & Collins, K. D. Specific cleavage of peptides containing an aspartic acid (β -hydroxamic acid) residue. *J. Am. Chem. Soc.* **107**, 4305–4313 (1985).
2. Kwong, M. Y. & Harris, R. J. Identification of succinimide sites in proteins by N-terminal sequence analysis after alkaline hydroxylamine cleavage. *Protein Sci.* **3**, 147–9 (1994).
3. Zhu, J. X. & Aswad, D. W. Selective cleavage of isoaspartyl peptide bonds by hydroxylamine after methyltransferase priming. *Anal. Biochem.* **364**, 1–7 (2007)
4. Ouellette, D., Chumsae, C., Clabbers, A., Radziejewski, C. & Correia, I. Comparison of the in vitro and in vivo stability of a succinimide intermediate observed on a therapeutic IgG1 molecule. *MAbs* **5**, 432–44
5. Ni, W., Dai, S., Karger, B. L. & Zhou, Z. S. Analysis of isoaspartic Acid by selective proteolysis with Asp-N and electron transfer dissociation mass spectrometry. *Anal. Chem.* **82**, 7485–91 (2010).
6. Liu, S., Moulton, K. R., Auclair, J. R. & Zhou, Z. S. Mildly acidic conditions eliminate deamidation artifact during proteolysis: digestion with endoprotease Glu-C at pH 4.5. *Amino Acids* **48**, 1059–67 (2016)
7. Sargaeva, N. P., Lin, C. & O'Connor, P. B. Unusual fragmentation of β -linked peptides by ExD tandem mass spectrometry. *J. Am. Soc. Mass Spectrom.* **22**, 480–91 (2011).

Reviewer 3

Summary of the key results

This paper describes an intriguing finding of a remarkably stable succinimide in an enzyme from a hyperthermophilic archaeon. This is contrary to the belief that succinimide is transiently formed as an intermediate during asparaginyl deamidation or aspartyl dehydration in proteins, and its formation is followed by rapid hydrolysis of this intermediate to aspartyl and isoaspartyl residue.

A significant part of the evidence is based on mass spectrometric data presented in Figures 1, 2, 4, and 5, as well as Supplementary Figures 1, 2, 3, 4, 6 and 7. As this reviewer is an expert in mass spectrometry, these data drew most of his attention. The mass spectrometry data please with their abundance, but are not without flaws. These potential flaws are discussed below (not necessarily in the order of significance).

Response to reviewer 3

#1. First off, the protein spectra (Fig. 1b, e, f; 2 a, b, e, f; etc.) are all taken with different signal-to-noise ratios, ranging from ≈ 5 -6 in Fig. 1e to ≈ 20 in Fig. 2b. This could mean that the spectra were taken at different source conditions (temperature, nozzle-skimmer voltage, source cleanliness, etc.). Often, spectra taken at lower S/N show a single peak (e.g., Fig. 1e, 2a), or a dominant peak at a large m/z (Fig. 5c), while those taken at higher S/N show either plurality of peaks (e.g., 1f, 2b and f), or dominant peaks at lower m/z (e.g., 2e, 5b). Could that be an artifact of different source conditions? It is well known in mass spectrometry, that depending upon source conditions, one can obtain protein spectra with different extent of small-molecule losses, such as NH_3 and H_2O loss, i.e., -17 and -18 Da, respectively. These artificial losses would be indistinguishable in mass from the mass defect due to the succinimide presence.

Response

Under the ESI conditions of gas temperature and capillary voltage used, neutral losses of NH_3 (-17 Da) and H_2O (-18Da) are not observed for a wide variety of proteins examined. The complete absence of succinimide in MjGATase_N109S (**Figure 1e**) in the ESI-MS recorded with similar parameters used for wild-type MjGATase suggests that the presence of succinimide in WT MjGATase is not a consequence of artifacts arising from the source conditions.

Besides checking the reproducibility of the mass differences under Q-TOF conditions, we have also recorded the ESI spectra on Orbitrap mass spectrometer to rule out experimental artifacts, if any. The ESI-MS of WT MjGATase and MjGATase_N109 recorded on an Orbitrap also shows loss of 17 Da from the WT MjGATase while the mass of MjGATase_N109S is in complete agreement with mass expected from the sequence of this protein (**Supplementary Fig.3**). Loss of 17 Da in MjGATase is clearly a function of a post-translation modification of a specific asparaginyl residue (N109) to succinimide and not an experimental artifact. This has been extensively established by experiments (site-directed mutagenesis and biochemical studies) reported in this manuscript.

2. Adding to the above suspicion, the insets are shown with different magnification of the m/z scale, and the latter is never given. The difference in scales (e.g., compare insets in Fig. 2a and 2b) is probably one order of magnitude. Why would so different scales be needed? Even Supplementary Figure 3 shows different and unspecified scales.

Response

Insets to all the figures now have the scale specified in the revised manuscript.

#3. Contrary to what is customary in protein mass spectrometry, only one charge state is zoomed in, instead of demonstrating the results of neutral mass deconvolution. That is not surprising if the peak ratios change dramatically with the charge state, as would be in the case of gas-phase losses (artifacts), which would be stronger from higher charge states.

Response

In place of the most abundant charge state, result of neutral mass deconvolution has been included in the figures of the revised draft.

4. In general, the mass spectrometric resolution is quite low. Many of today's instruments, including qTOFs, Orbitraps and FT ICR MS can easily resolve isotopic peaks of a 23 kDa protein. With such isotopic resolution, the difference between the NH₃ and H₂O losses would be much more clear. Why hasn't been high resolution used in at least most important cases?

Response

We agree with the reviewer that many of the present day Q-TOFs, Orbitraps and especially FTICR yield data with unprecedented resolution. However, we believe our mass spectral data are of sufficient resolution to support our claims. Furthermore, we have validated our mass spectral findings by site-directed mutagenesis and biochemical studies. ESI-MS of intact protein, MALDI, CID and ETDMS/MS of tryptic peptides of MjGATase and site-directed mutagenesis, taken together provide very strong support for succinimide formation at position N109 in MjGATase. Further, as suggested by this reviewer we have now acquired ESI-MS on Q Exactive hybrid quadrupole-Orbitrap mass spectrometer at 70,000 resolution. This also shows loss of 17 Da in complete agreement with data obtained on a Q-TOF mass spectrometer (**Supplementary Figure 3**). These data have been included in the revised draft.

5. It is great that the authors used a plurality of MS/MS techniques, including CID and ETD, but the results are not always consistent with their conclusions. It is amazing that the peptide in Supplementary Fig. 1a has survived harsh MALDI conditions while preserving its succinimide. Equally remarkable is that this peptide has apparently survived also electrospray ionization to produce an ETD MS/MS spectrum in Fig. 1c. The authors don't comment on this apparent stability of succinimide in a peptide. If it is so stable, why not perform LC-MS/MS analysis and quantify peptide abundances as common in proteomics instead of relying on low-resolution protein mass spectra?

Response

The referee has expressed surprise in the tryptic fragment retaining succinimide under both ESI and MALDI conditions. It should be noted that the succinimidyl derivative in the tryptic peptide is present only in a relatively minor amount (**Supplementary Figure 2a**), being more susceptible to hydrolysis than in intact protein (**Figure 1b**) However, the stability of the imide intermediate under mass spectral conditions is not really surprising as succinimide intermediates are stable to both soft ionization, MALDI and ESI.

6. In all MS/MS spectra, the C-terminal series of fragments (y- and z-) appear to be mislabeled, with a series starting from z2 or y2 ions that have only one amino acid, K.

Response

Mislabeled y and z ions are corrected in the revised draft. We apologize for the error.

References

1. Vieille, C. & Zeikus, G. J. Hyperthermophilic enzymes: sources, uses, and molecular mechanisms for thermostability. *Microbiol. Mol. Biol. Rev.* **65**, 1–43 (2001).
2. Blodgett, J. K., Loudon, G. M. & Collins, K. D. Specific cleavage of peptides containing an aspartic acid (β -hydroxamic acid) residue. *J. Am. Chem. Soc.* **107**, 4305–4313 (1985).
3. Kwong, M. Y. & Harris, R. J. Identification of succinimide sites in proteins by N-terminal sequence analysis after alkaline hydroxylamine cleavage. *Protein Sci.* **3**, 147–9 (1994).
4. Zhu, J. X. & Aswad, D. W. Selective cleavage of isoaspartyl peptide bonds by hydroxylamine after methyltransferase priming. *Anal. Biochem.* **364**, 1–7 (2007).
5. Reissner, K. J. & Aswad, D. W. Deamidation and isoaspartate formation in proteins: unwanted alterations or surreptitious signals? *Cell Mol Life Sci* **60**, 1281–1295 (2003).
6. Aki, K., Fujii, N. & Fujii, N. Kinetics of isomerization and inversion of aspartate 58 of α A-crystallin peptide mimics under physiological conditions. *PLoS One* **8**, e58515 (2013).

Reviewers' Comments:

Reviewer #1 (Remarks to the Author)

Kumar et al. have carefully answered to all points I had raised on the previous version of their manuscript.

Moreover, the new version describes new experiments in response to Reviewer's 2 comments (stability of the succinimide in the presence of 0.1M HCl and 2M NH₂OH, stability of a synthetic peptide) that remarkably complete the previous version of the manuscript. I strongly recommend Nature Communication to publish this version of the manuscript.

Correction

Legend of Figure 2 is wrong. Correct to:

(b) Scheme showing fragmentation by CID and ETD. Under CID

conditions, peptide bond cleavage predominates, leading to the generation of c and z ions.

Fragmentation by ETD leads to cleavage between C α and C β , which generates the c+57 and z-57 ions.

Reviewer #2 (Remarks to the Author)

The authors have done a good job addressing all the issues.

Reviewer #3 (Remarks to the Author)

I am in general satisfied with the amendments the Authors have made, and with their replies.